# Dependence of myosin filament structure on intracellular calcium concentration in skeletal muscle

Marco Caremani[1]* , Luca Fusi[2,3]* , Massimo Reconditi[1,4] , Gabriella Piazzesi[1] , Theyencheri Narayanan[5] , Malcolm Irving[2] , Vincenzo Lombardi[1] , Marco Linari[1,4]** , and Elisabetta Brunello[2]**

**Contraction of skeletal muscle is triggered by an increase in intracellular calcium concentration that relieves the structural block on actin-binding sites in resting muscle, potentially allowing myosin motors to bind and generate force. However, most myosin motors are not available for actin binding because they are stabilized in folded helical tracks on the surface of myosin-containing thick filaments. High-force contraction depends on the release of the folded motors, which can be triggered by stress in the thick filament backbone, but additional mechanisms may link the activation of the thick filaments to that of the thin filaments or to intracellular calcium concentration. Here, we used x-ray diffraction in combination with temperature-jump activation to determine the steady-state calcium dependence of thick filament structure and myosin motor conformation in near-physiological conditions. We found that x-ray signals associated with the perpendicular motors characteristic of isometric force generation had almost the same calcium sensitivity as force, but x-ray signals associated with perturbations in the folded myosin helix had a much higher calcium sensitivity. Moreover, a new population of myosin motors with a longer axial periodicity became prominent at low levels of calcium activation and may represent an intermediate regulatory state of the myosin motors in the physiological pathway of filament activation.**

## Introduction

Muscle contraction is driven by relative sliding between myosin-containing thick filaments and actin-containing thin filaments. Myosin head or motor domains anchored by their tails in the thick filaments bind to actin, change shape to drive filament sliding, and then detach in a mechano–chemical cycle coupled to ATP hydrolysis (Huxley, 1969; Huxley and Simmons, 1971; Lymn and Taylor, 1971). Muscle contraction is initiated by an intracellular calcium transient; calcium ions bind to troponin in the thin filaments, causing a structural change that exposes the myosin binding sites on actin (Huxley, 1973; Gordon et al., 2000). Many of the myosin motors in the resting muscle are not available for actin binding, however, because they are folded back against their tails in a helical array on the surface of the thick filaments stabilized by multiple interactions between myosin motors, their tails, titin, and myosin-binding protein-C (Huxley and Brown, 1967; Woodhead et al., 2005; Zoghbi et al., 2008; Reconditi et al., 2011; Al-Khayat et al., 2013; Dutta et al., 2023 *Preprint*; Tamborrini et al., 2023 *Preprint*). This folded helical state of the motors also inhibits their ATPase activity,

minimizing the metabolic cost of resting muscle (Stewart et al., 2010).

The helical array of folded myosin motors can be monitored in intact muscles because it gives rise to a characteristic x-ray reflection called the first myosin layer line (ML1; Huxley and Brown, 1967). Time-resolved x-ray measurements of the intensity of the ML1 layer line ($I_{ML1}$) in whole muscles (Huxley et al., 1982; Hill et al., 2021) and isolated muscle fibers (Piazzesi et al., 1999; Reconditi et al., 2011) showed that myosin motors leave the folded helical state following electrical stimulation not as quickly as activation of the thin filament (Kress et al., 1986), but faster than the attachment of motors to actin and force generation. These results suggest that activation of the thin filament in response to calcium binding is rapidly signaled to the thick filament to release myosin motors from the folded helical state.

Subsequent x-ray studies showed that this interfilament signaling mechanism is partly mechanical; it can be inhibited or delayed by imposing rapid shortening soon after electrical stimulation to reduce the stress in the filaments to near zero

......................................................................................................................................................................................................................

[1]PhysioLab, University of Florence, Florence, Italy;   [2]Randall Centre for Cell and Molecular Biophysics and British Heart Foundation Centre of Research Excellence, King's College London, London, UK;   [3]Centre for Human and Applied Physiological Sciences, King's College London, London, UK;   [4]Consorzio Nazionale Interuniversitario per le Scienze Fisiche della Materia, Florence, Italy;   [5]European Synchrotron Radiation Facility, Grenoble, France.

*M. Caremani and L. Fusi contributed equally to this paper;   **M. Linari and E. Brunello contributed equally to this paper.   Correspondence to Elisabetta Brunello: elisabetta.brunello@kcl.ac.uk.

(Linari et al., 2015; Hill et al., 2022), implying that myosin motors are released from the folded helical state when stress is applied to the thick filament backbone. A small population of motors that are not folded in resting muscle might be available for binding to actin when the thin filament is activated, and these "constitutively on" or "sentinel" motors could generate enough active force to activate the folded helical motors through thick filament mechanosensing (Linari et al., 2015; Irving, 2017; Piazzesi et al., 2018; Craig and Padrón, 2022). The molecular structural basis of thick filament mechanosensing is unknown, although the thick filament backbone has a slightly longer axial periodicity in contracting muscle, and this periodicity can also be monitored precisely using the spacing of the sixth-order myosin-based meridional x-ray reflection ($S_{M6}$; Reconditi et al., 2004; Huxley et al., 2006). $S_{M6}$ increases following electrical stimulation with about the same time-course as the release of myosin motors from the folded helical state (Reconditi et al., 2011; Hill et al., 2021), and this increase is greatly attenuated by the imposition of rapid shortening to remove filament stress (Linari et al., 2015; Hill et al., 2022). These results suggest that filament stress alters the packing of the myosin tails in the filament backbone, leading to an arrangement with a slightly longer axial periodicity that is somehow linked to the reduced affinity of the motors for the backbone.

Mechanosensing may not be the only mechanism of thick filament activation, however. Recent evidence suggests that thick filaments may be directly regulated by calcium in heart muscle (Ma et al., 2022), and that the I-band region of titin becomes much stiffer on activation in skeletal muscle, potentially coupling a calcium-dependent interaction of titin with the thin filament in the I band to a titin-mediated change in the structure of the thick filaments in the A band (Squarci et al., 2023). To better understand and constrain possible mechanisms that might link thick filament activation to intracellular free calcium concentration ($[Ca^{2+}]$) in skeletal muscle, here we used x-ray diffraction to determine the dependence of thick filament structure on $[Ca^{2+}]$ in the steady state. Steady-state calcium titrations of this type have been used extensively to investigate the regulation of muscle contraction in demembranated muscle fibers (Brenner and Yu, 1985; Gordon et al., 2000). The dependence of isometric force on $[Ca^{2+}]$, expressed as $pCa = -\log_{10}[Ca^{2+}]$, is generally fitted using the Hill equation with parameters $pCa_{50}$, the value of pCa corresponding to half-maximal activation, and $n_H$, the Hill coefficient describing the steepness of the calcium dependence. This type of study generally used mammalian muscle fibers at relatively low temperatures to minimize the irreversible effects of maintained activation of such fibers at temperatures closer to the physiological. However, a major limitation of those studies, in general not recognized at the time of their publication, is that the thick filaments are already switched on in the absence of $[Ca^{2+}]$ at those low temperatures (Caremani et al., 2019, 2021); thick filament activation pathways were inadvertently excluded by the experimental design.

The folded state of the myosin motors can be preserved in demembranated fibers from mammalian skeletal muscle at a temperature of at least 25°C when the filament lattice is compressed by 5% dextran (Fusi et al., 2015; Caremani et al., 2021).

No previous x-ray study, to our knowledge, has described the calcium dependence of myosin conformation and thick filament structure during active contraction of skeletal muscle in these conditions. Here, we used a temperature-jump activation protocol (Linari et al., 2007) in bundles of demembranated fibers from rabbit psoas muscle that had been equilibrated in activating solution at 1°C to remove the inhomogeneity of muscle activation associated with slow diffusion of calcium into the core of the bundle and to minimize the duration at which the fibers were contracting at 25°C and the associated irreversible effects of activation at that temperature. We measured the force and x-ray reflections associated with the thick filaments and myosin motors at different levels of calcium activation in the steady state, with the general aim of constraining possible mechanisms of thick filament activation. We found that x-ray signals associated with the perpendicular motors characteristic of isometric force generation had about the same calcium sensitivity as force, but those associated with perturbations in the folded myosin helix had much higher calcium sensitivity and cooperativity. Moreover, a new population of myosin motors with a longer axial periodicity became prominent at low levels of calcium activation, which may represent an intermediate regulatory state of the myosin motors in the physiological pathway of filament activation.

## Materials and methods
### Fiber preparation
Glycerinated demembranated fiber segments from psoas muscles of adult male New Zealand white rabbits were prepared as described previously (Linari et al., 2007). Briefly, rabbits were killed by injection of an overdose of sodium pentobarbitone (150 mg kg$^{-1}$) in the marginal ear vein, in accordance with the official regulations of the Community Council (Directive 86/609/EEC) and with Schedule 1 of the UK Animals (Scientific Procedures) Act 1986. The study was approved by the Ethical Committee for Animal Experiments of the University of Florence. Bundles of 70–150 fibers were stored in a solution containing 50% glycerol at –20°C for 1–2 wk, and smaller bundles of fibers, 5–6 mm long, about 3–4 fibers wide, and 1–2 fibers deep, were dissected on the day of the experiment. T-shaped aluminum clips were used to attach the fiber ends to coaxial hooks, one of which was attached to a strain gauge force transducer (AE801; Sensonor). Before each experiment, the ends of the bundle were fixed with glutaraldehyde and glued to the clips with shellac dissolved in ethanol. The bundle was mounted horizontally in a relaxing solution at ~2.4 µm sarcomere length in a multidrop apparatus that allowed activation using a temperature jump technique (Linari et al., 2007; Fig. S1). The bundle was kept in a preactivating solution at a low temperature (1°C) for 2 min and then transferred to an activating solution at 1°C in which little force was developed. When force became steady (within 10 s), the bundle was transferred to activating solution at 25°C and, following full force development, it was transferred to air for the x-ray exposure (about 1 s after the temperature jump) to minimize scattering from the solution and the windows of the chamber.

Force, motor position, and x-ray acquisition timing were collected and analyzed using LabVIEW (National Instruments).

## Fluorescence polarization experiments

Single demembranated rabbit muscle fibers were mounted in a relaxing solution on the mechanical apparatus of a fluorescence polarization setup (Fusi et al., 2014), similar to that of the x-ray diffraction setup described above, and fiber extremities were fixed with glutaraldehyde and glued to aluminum clips with shellac dissolved in ethanol. The native RLC in the muscle fiber was partially (~30%) replaced with a mutant RLC in which the E helix was bifunctionally conjugated to rhodamine, using an incubation in EDTA-rigor buffer for 40′ at 19°C containing ~20 µM of the labeled RLC, as described previously (Fusi et al., 2015). The exchanged fibers were activated by a temperature jump from 1 to 25°C. In the $Ca^{2+}$-titration experiments, force and the polarized fluorescence from the RLC probe were measured in solution after the force reached the steady state (about 0.5 s after the temperature jump at pCa 4.7 and up to 5 s after the temperature jump at low levels of calcium activation), and the order parameter of the probe, $<P_2>$, was calculated as previously described (Fusi et al., 2015).

## Solutions

The composition of the solutions used in the x-ray experiments was as described by Linari et al. (2007). The skinning solution contained 10 mM imidazole, 2.5 mM $MgCl_2$, 5 mM ethylene glycol-bis-(β-aminoethyl ether)-N,N,N9,N9-tetraacetic acid (EGTA), 2.5 mM $Na_2ATP$, 170 mM potassium propionate (KP), and 0.1 mM phenylmethylsulphonyl fluoride (PMSF). The storage solution contained 10 mM imidazole, 2.5 mM $MgCl_2$, 5 mM EGTA, 2.5 mM $Na_2ATP$, 170 mM KP, 5 mM $NaN_3$, and 50% glycerol (vol/vol). The relaxing solution contained 100 mM N-tris[hydroxymethyl]methyl-2-aminoethanesulphonic acid (TES) buffer, 7.7 mM $MgCl_2$, 25 mM EGTA, 5.4 mM $Na_2ATP$, 19.1 mM $Na_2$-creatine phosphate (CP), and 10 mM reduced glutathione (GSH). A preactivating solution contained 100 mM TES, 6.9 mM $MgCl_2$, 0.1 mM EGTA, 24.9 mM HDTA, 5.5 mM $Na_2ATP$, 19.5 mM $Na_2CP$, and 10 mM GSH. The activating solution contained 100 mM TES, 6.8 mM $MgCl_2$, 25 mM CaEGTA, 5.5 mM $Na_2ATP$, 19.5 mM $Na_2CP$, and 10 mM GSH. All solutions had 5 mM Mg-ATP; 1.2 mM free $Mg^{2+}$; 199 mM ionic strength; pH 7.1 at 25°C. The composition of the solutions used in the fluorescence polarization experiments was as described by Fusi et al. (2015); main differences with respect to solutions used for x-ray experiments are lower ionic strength (150 versus 199 mM), a different pH buffer (25 mM imidazole versus 100 mM TES) and a lower concentration of EGTA/CaEGTA (10 versus 25 mM). The pH of all solutions was adjusted to 7.1 at 25°C. Relaxing and activating solutions were mixed to obtain a series of partial activating solutions with the required free calcium ion concentrations, calculated using software kindly provided by Prof. Earl Homsher (University of California Los Angeles, Los Angeles, CA, USA). The osmotic agent dextran T500 (5% wt/vol; Pharmacia Biotech) was added to the experimental solutions to reduce the interfilament spacing in relaxing conditions to a value about 1 nm lower than that in resting intact EDL muscle of the mouse (Linari et al., 2007; Caremani et al., 2019, 2021).

## X-ray data collection

The fibers were horizontally mounted in the thermoregulated mechanical apparatus described above mounted on an XYZ stage (Fig. S1). The Z movement allowed the plate carrying the solution drops to be dropped so that the fiber could be irradiated by x-rays in a narrow aluminum-delimited air chamber to avoid x-ray absorption by the solution drops. The temperature of the air chamber was the same as the test temperature. When the x-ray exposure was recorded in relaxing conditions, the temperature of all the drops was the same and the bundle was transferred to the air chamber limiting the number of meniscus interfaces the bundle had to pass.

The apparatus was mounted at beamline ID02 of the European Synchrotron Radiation Facility (ESRF). The beam was slit collimated to a size 120 µm × 30 µm (horizontal × vertical, full width at half-maximum) at the detector position which corresponded to $2.3 \times 10^{12}$ photons $s^{-1}$ at 0.1 nm wavelength. The beam was attenuated for bundle alignment. To minimize radiation damage, x-ray exposure was limited to the data collection period using a fast electromagnetic shutter (nmLaser Products, Inc.) and the bundle was moved horizontally by 200–400 µm between exposures. 20-ms time frames were collected at each calcium concentration. Total exposure time varied from 300 to 440 ms in each bundle before radiation damage occurred. X-ray diffraction patterns were recorded using a Rayonix MX 170 HS detector, which has an active area of 170 mm × 170 mm and a pixel size of 44.2 µm × 44.2 µm (Narayanan et al., 2018). The 3,840 × 3,840 pixels of the detector were binned by a factor of 2 in both the horizontal and vertical direction before readout to increase the signal-to-noise ratio on the weaker reflections. The camera length was set to 5 m to record the interference fine structure in the meridional reflections up to the sixth order of the myosin reflections on both sides of the x-ray pattern. X-ray data are presented from four bundles of fibers of length 2,870 ± 280 µm, cross-sectional area 28,460 ± 2,130 µm²; initial sarcomere length, 2.41 ± 0.04 µm; force at saturating calcium at 25°C, 239.4 ± 17.0 kPa (mean ± SE).

## Data analysis

X-ray diffraction data were analyzed using the SAXS package (P. Boesecke; ESRF), Fit2D (A. Hammersley; ESRF), and IgorPro (WaveMetrix, Inc.). 2-D patterns were centered and aligned using the equatorial 1,0 or 1,1 reflections, then mirrored horizontally and vertically. The equatorial intensity distribution was determined from each 2-D x-ray diffraction pattern of the four bundles by integrating from 0.0036 $nm^{-1}$ on either side of the equatorial axis, and the intensities of the 1,0 and 1,1 reflections were determined by fitting four Gaussian peaks (1,0; Z line; 1,1; and 2,0) with spacing multiples of the 1,0 position. $d_{11} = d_{10}/(3)^{1/2}$; $d_{20} = d_{10}/2$; and $d_Z = d_{10} \times 0.682$. The distribution of diffracted intensity along the meridional axis of the x-ray pattern (parallel to the fiber axis) was calculated by integrating from 0.012 $nm^{-1}$ on either side of the meridian for the myosin-based M1, M2, M3, and M6 reflections, and troponin T1 reflection. The

first myosin layer line (ML1) was integrated in the region between 0.018 and 0.076 nm$^{-1}$ from the meridional axis. The integration limits were chosen to collect the whole intensity of the ML1 reflection corresponding to the 1,0 and 1,1 row lines (Caremani et al., 2021); the width of the M1 meridional reflection was much smaller than the selected integration limits.

The 1-D diffracted intensity profiles of the meridional and layer line reflections for each bundle were divided by their respective $I_{10}$ values at low calcium concentration (pCa 9, 25°C) to control for variations of the mass in the beam. Background intensity distributions were fitted using a convex hull algorithm and subtracted; the small background remaining when the convex hull algorithm had been used was removed using the intensity from a nearby region of the x-ray pattern containing no reflections or with a linear fit. Integrated intensities were obtained from the following axial regions: M1, 0.021–0.024 nm$^{-1}$; T1, 0.025–0.028 nm$^{-1}$; M2L (low angle component of the M2), 0.040–0.045 nm$^{-1}$; M2H (high angle component of the M2), 0.045–0.049 nm$^{-1}$; M3, 0.064–0.073 nm$^{-1}$; and M6, 0.133–0.144 nm$^{-1}$. The cross-meridional width of the M3 and M6 reflections was determined from the integrated intensity in a zone parallel to the equatorial axis in the axial regions specified above for the two reflections and a double Gaussian fit centered on the meridian with the narrower Gaussian in the region ±0.030 nm$^{-1}$. The interference components of the M1, M2L, M2H, M3, and M6 reflections were determined by fitting multiple Gaussian peaks with the same axial width to the meridional intensity distribution, and the total intensity of the reflection was calculated as the sum of the component peaks. The spacing of each reflection was determined from the weighted mean of the component peaks and calibrated using as a reference the spacing of 14.358 nm measured from psoas muscle fibers at steady-state relaxation at pCa 9 at 25°C (Caremani et al., 2021).

Hill curve fittings {y = B + A*[1/(1 + 10$^{(nH*(pCa − pCa50))}$)]} were performed on force and x-ray parameters from each bundle obtained with the T-jump protocol (pCa range 7–4.5). The Hill curves shown in Figs. 1 D; 2, B and C; 3, B and C; 6, B and C; 7; 8; and S3 were calculated from the mean $n_H$ and $pCa_{50}$ reported in Table 1, and the mean values of the parameters A and B. The low signal-to-noise of the M6 reflection prevented bundle-by-bundle analysis, so the 1-D profiles from the four bundles were averaged before fitting the reflection profile. The Hill curve in Fig. 5 is therefore a fit to these averaged data, and the SD for $S_{M6}$ in Table 1 was obtained from the fitting program.

### Statistical analysis
Data presented are mean ± SD or SE as indicated in the text and in the figure or table legends. Paired Student $t$ tests were performed to compare values for $n_H$ and $pCa_{50}$ between force and the x-ray structural parameters; P values are reported in Table 1.

### Online supplemental material
Fig. S1 shows a diagram of the experimental setup. Fig. S2 reports the changes in the intensity of the M6 reflection as a function of pCa. Fig. S3 shows the intensity and spacing of the M3 reflection subpeaks and the star peak as a function of pCa.

## Results
### Comparison of force–calcium relationships in temperature-jump and steady-state activation protocols
We used small bundles of about six demembranated fibers from rabbit psoas muscle for the x-ray experiments to record the weaker x-ray reflections with adequate signal:noise, and we achieved the necessary uniform and rapid activation of these fiber bundles at different pCa values using a temperature jump from 1 to 25°C (Linari et al., 2007). To allow the results from that protocol to be compared with those obtained in force–pCa titrations using a more conventional protocol involving preincubation in a solution of low calcium buffering capacity before transfer to a highly buffered solution with the required free [Ca$^{2+}$], we made some control measurements in single demembranated fibers, in which the results from two protocols can be directly compared. The relationship between active force and free [Ca$^{2+}$] in the conventional protocol in the steady state at 25°C in the presence of 5% dextran, expressed on a logarithmic scale as pCa = −log$_{10}$ [Ca$^{2+}$], was fitted with the Hill equation (Fig. 1 A, orange; Fusi et al., 2016). The steady force produced following rapid activation by temperature jump from 1 to 25°C (Fig. 1 A, black) was close to that obtained using the conventional preactivation protocol at each [Ca$^{2+}$] (orange). There was a small difference between the best-fit values of $pCa_{50}$, the pCa corresponding to half-maximum force, in the two protocols but no significant difference in the Hill coefficient $n_H$ that measures the steepness of the calcium dependence (Table 1).

### Calcium dependence of myosin motor conformation in temperature-jump and steady-state activation protocols
The change in the conformation of the myosin motors in single demembranated muscle fibers was measured in the temperature jump protocol using polarized fluorescence from a bifunctional rhodamine probe attached along the E helix of the myosin regulatory light chain (RLC-E; Fusi et al., 2016). The measured order parameter <$P_2$> for this probe decreases on activation as myosin motors leave the folded helical conformation and the average orientation of the probe dipoles becomes less parallel to the fiber axis. The <$P_2$> values observed on activation by temperature jump to 25°C (Fig. 1 B, black) were fitted with the Hill equation with $pCa_{50}$ 6.49 ± 0.01 and $n_H$ 6.94 ± 0.87 (mean ± SD), similar to the published steady-state values obtained with the conventional protocol at that temperature, $pCa_{50}$ 6.53 ± 0.01 and $n_H$ 5.80 ± 0.17 (Fusi et al., 2016; Fig. 1 B, orange; Table 1). The conformation of myosin motors at maximal calcium activation as determined by <$P_2$> for the RLC-E probe was also similar in the two protocols (Fig. 1 B), but those at calcium concentrations that were subthreshold for activation were significantly different (for example P = 0.015 for pCa 9, $t$-test).

The results in Fig. 1 B show that the folded motor conformation observed in relaxing conditions in the steady state at 25°C, which is lost on cooling to 1°C, was not fully recovered after a temperature jump from 1 to 25°C. Time-resolved measurements of <$P_2$> from the RLC-E probe following such a temperature jump in relaxing conditions (pCa 9) revealed a rapid change in the first 100 ms (Fig. 1 C, black), but also a much slower component with a time constant of several seconds, which accounts for the deficit in <$P_2$> observed in the titration in

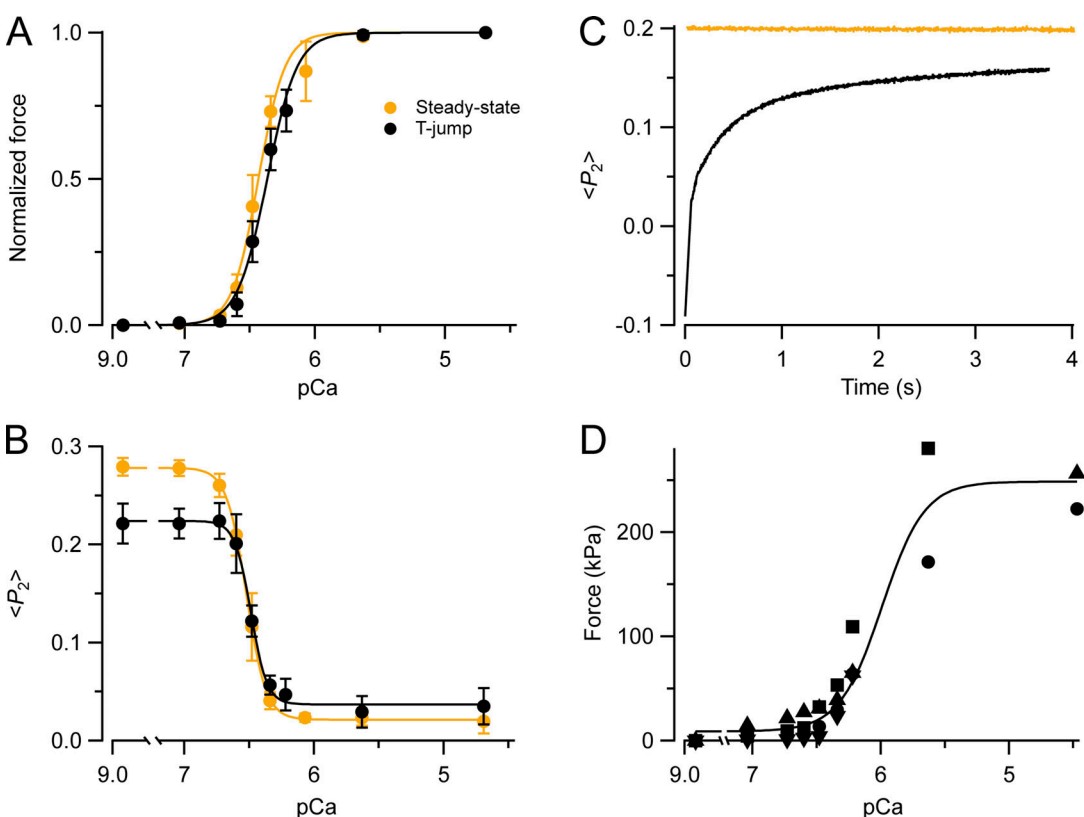

Figure 1. **Dependence of force and RLC orientation on calcium concentration in the steady state. (A)** Force at different pCa normalized by the maximum force at pCa 4.7, obtained with a conventional preactivation protocol at 25°C (orange circles, $T_0$ = 267 ± 10 kPa; mean ± SE, $n$ = 5 fibers; data from Fusi et al. [2016]) or using the temperature jump protocol (black circles; $T_0$ = 286 ± 25 kPa; mean ± SE, $n$ = 3 fibers). Data from single psoas muscle fibers; sarcomere length, 2.40 μm; temperature, 25°C; 5% dextran. **(B)** <$P_2$> for the RLC E-helix probe in the same experiments as A. Lines in A and B are fits of the Hill equation to the average <$P_2$> at each pCa. **(C)** <$P_2$> for the RLC E-helix probe at pCa 9, 25°C in the steady state (orange line) and after a temperature jump to 25°C (black line) at time zero. **(D)** Force at different pCa following temperature jumps to 25°C; data points from individual fiber bundles are identified by different symbols. The line represents the Hill equation for the mean parameters in Table 1 obtained from four fiber bundles.

Fig. 1 B (black) compared with the steady state in the same conditions (Fig. 1, B and C, orange).

## Force–calcium relationships in the fiber bundles used for the x-ray experiments

In the x-ray experiments, force and x-ray patterns were recorded in bundles of muscle fibers after a temperature jump to 25°C followed by the removal of the solution to minimize absorption and scattering during the x-ray exposure. Force at maximal calcium activation (pCa 4.5) was 239.4 ± 17.0 kPa, similar to that measured at the same temperature at the plateau of an isometric tetanus in intact EDL muscles of the mouse (Caremani et al., 2019). Force values at different pCa were fitted with the Hill equation (Fig. 1 D, continuous line). $pCa_{50}$ was 5.99 ± 0.17 (mean ± SD, $n$ = 4) and $n_H$ was 2.63 ± 1.43 (Table 1). The lower $pCa_{50}$ and $n_H$ values for force in the x-ray experiments compared with those with the RLC-E probe experiments are likely to be associated with the use of fiber bundles contracting in air at the time of the x-ray exposure (see Discussion).

## Equatorial x-ray reflections

The intensity distribution on the equator of the x-ray diffraction pattern (Fig. 2 A) is sensitive to the distribution of mass projected onto a plane perpendicular to the thick and thin filaments, and therefore to radial movements of the myosin motors with respect to their thick filament origins. In relaxing conditions at 25°C (red), the 1,0 reflection is brighter than the 1,1 reflection, indicating that the myosin motors are close to the thick filament surface, as expected for the folded helical conformation (Caremani et al., 2019, 2021). As the calcium concentration is increased, the 1,0 reflection becomes weaker and the 1,1 stronger, indicating that motors leave the folded state and move closer to the thin filaments. The ratio of the 1,1 and 1,0 intensities, $I_{11}/I_{10}$ (Fig. 2 B), a convenient metric for this movement, increased from 0.6 in relaxing conditions to 1.95 at full calcium activation (pCa 4.5). These values are close to those observed in intact mouse EDL muscles at the same temperature at rest and at the plateau of isometric tetanus, respectively (Caremani et al., 2019).

In Fig. 2 B, as in subsequent calcium titration figures, we have expanded the x axis to show more clearly the data obtained with the temperature jump protocol at pCa 7 and below. The force obtained in the same protocol is shown as the black line, which is from the Hill equation fit to force in Fig. 1 D. The horizontal magenta line indicates the steady-state value of the same x-ray parameter during steady-state relaxation at pCa 9, respectively.

**Table 1. Calcium dependence of force and structural parameters**

|  | $n_H$ | $pCa_{50}$ |
|---|---|---|
| Force (fiber bundles) | 2.63 ± 1.43 | 5.99 ± 0.17 |
| $I_{11}/I_{10}$ | 4.72 ± 1.52 (P = 0.096) | 6.18 ± 0.05 (P = 0.089) |
| $d_{10}$ | 2.88 ± 0.97 (P = 0.422) | 6.40 ± 0.10 (P = 0.013) |
| $I_{ML1}$ | 3.53 ± 1.83 (P = 0.265) | 6.56 ± 0.14 (P = 0.004) |
| $A_{ML1}$ | 2.93 ± 1.94 (P = 0.430) | 6.49 ± 0.10 (P = 0.007) |
| $S_{M6}$[a] | 2.58 ± 0.48[a] | 6.18 ± 0.03[a] |
| $S_{M3}$ | 5.11 ± 1.37 (P = 0.037) | 6.24 ± 0.02 (P = 0.041) |
| $I_{M3}$ | 4.34 ± 2.09 (P = 0.046) | 6.07 ± 0.09 (P = 0.251) |
| $A_{M3}$ | 4.17 ± 1.77 (P = 0.064) | 6.13 ± 0.07 (P = 0.144) |
| $L_{M3}$ | 3.45 ± 1.52 (P = 0.205) | 6.22 ± 0.03 (P = 0.039) |
| $I_{M1}$ | 4.25 ± 1.05 (P = 0.119) | 6.55 ± 0.06 (P = 0.005) |
| $I_{M2H}$ | 4.10 ± 1.63 (P = 0.038) | 6.65 ± 0.06 (P = 0.003) |
| $I_{M2L}$ | 3.60 ± 1.30 (P = 0.117) | 6.63 ± 0.09 (P = 0.008) |
| $A_{M2H}$ | 3.88 ± 1.85 (P = 0.044) | 6.59 ± 0.04 (P = 0.003) |
| $A_{M2L}$ | 3.14 ± 1.91 (P = 0.221) | 6.54 ± 0.07 (P = 0.006) |
| $I_{T1}$ | 6.12 ± 1.65 (P = 0.009) | 6.61 ± 0.08 (P = 0.007) |
| $A_{T1}$ | 7.00 ± 1.11 (P = 0.003) | 6.59 ± 0.09 (P = 0.008) |
| Force$_{SS}$ (single fibers) | 4.31 ± 0.55 | 6.43 ± 0.01 |
| $<P_2>_{SS}$ (single fibers) | 5.80 ± 0.17 | 6.53 ± 0.01 |
| Force$_{Tj}$ (single fibers) | 3.82 ± 0.32 (P = 0.935)[b] | 6.37 ± 0.01 (P = 0.157)[b] |
| $<P_2>_{Tj}$ (single fibers) | 6.94 ± 0.87 (P = 0.596)[b] | 6.49 ± 0.01 (P = 0.510)[b] |

Mean Hill equation parameters ($n_H$ and $pCa_{50}$) corresponding to the fits shown in Figs 1, 2, 3, 4, 5, 6, 7, and 8; and Fig. S3, with SD calculated from the fits of data from individual bundles. P values for $n_H$ and $pCa_{50}$ are the results of paired *t*-tests between force and x-ray structural parameters. Force and $<P_2>$ values from fluorescence polarization experiments on single fibers exchanged with an RLC E-helix probe were obtained using either a conventional preactivation protocol at 25°C (steady state, SS; data from Fusi et al. [2016]) or the temperature jump protocol (Tj).
[a]Due to lower signal-to-noise, Hill equation parameters for $S_{M6}$ were obtained from the average meridional intensity distributions for four bundles, and SD was obtained from the fitting program.
[b]P values from unpaired *t*-tests between force or $<P_2>$ in steady state and temperature jump protocols.

In general, this value is different from that measured following a temperature jump to pCa 7, as seen in the control experiments on single muscle fibers using polarized fluorescence from the RLC-E probes (Fig. 1 B), suggesting that the difference is due to incomplete recovery of the OFF state of the myosin motors following the temperature jump (see Discussion). Only the x-ray data obtained with the temperature-jump protocol were used for the fit of the x-ray data to the Hill equation so as to exclude the deficit in the structural changes associated with the slow recovery of the OFF state following the temperature jump (Fig. 1 C). The relationship between $I_{11}/I_{10}$ and pCa (Fig. 2B) was sigmoidal, and a fit to the Hill equation gave $pCa_{50}$ 6.18 ± 0.05 and $n_H$ 4.72 ± 1.52 (Table 1), showing that $I_{11}/I_{10}$ was more sensitive to calcium than force (black line). The spacing of the filament lattice ($d_{10}$, measured from the spacing of the 1,0 reflections) increased with

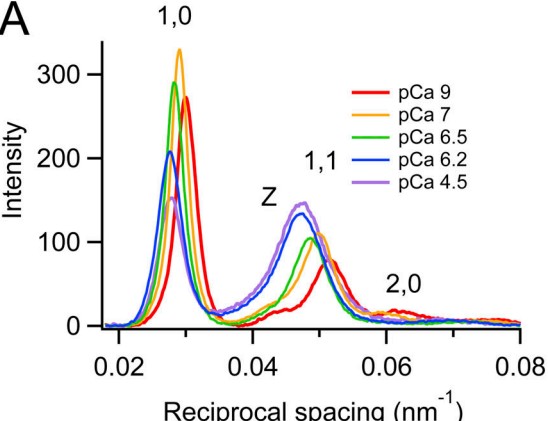

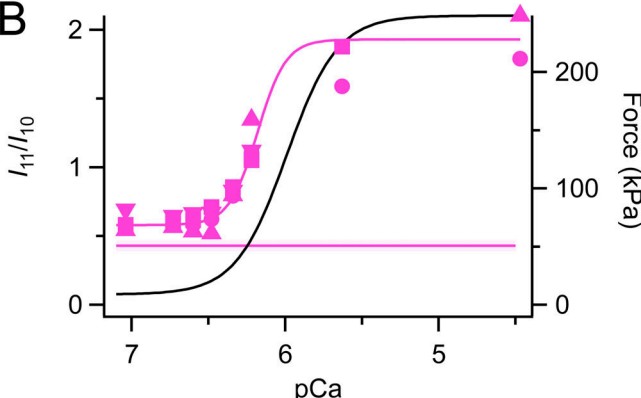

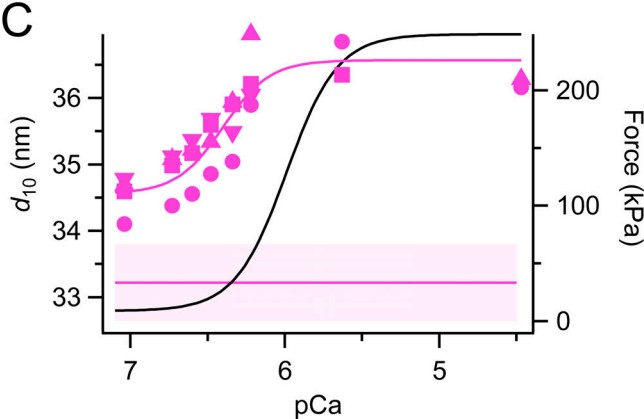

Figure 2. **Equatorial x-ray reflections. (A)** Equatorial intensity distributions at different calcium concentrations (average of two to four bundles). pCa 9, red; pCa 7, orange; pCa 6.5, green; pCa 6.2, blue; and pCa 4.5, violet. **(B and C)** Ratio of equatorial intensities ($I_{11}/I_{10}$, B) and spacing of 1,0 reflection ($d_{10}$, C) following a temperature jump to 25°C. Different magenta symbols identify individual fiber bundles. The magenta line represents the Hill equation for the mean parameters in Table 1 obtained from four fiber bundles. Horizontal magenta lines and shading (mean ± SD, n = 4 bundles) denote steady-state values at pCa 9. Black lines are the Hill fit to force from Fig. 1 D.

increasing [Ca²⁺] in the range pCa 7.0 to 4.5 after a temperature jump (Fig. 2 C). This increase is probably associated with the shortening of the central sarcomeres that are illuminated by the x-ray beam against the compliance of the end regions.

## Myosin-based helical order

The first myosin-based layer line (ML1) is associated with the folded helical arrangement of myosin motors on the surface of the thick filament (Huxley and Brown, 1967; Reconditi et al., 2011; Hill et al., 2021). It is strong in relaxed muscles at pCa 9 and 25°C, and its intensity, $I_{ML1}$, decreases with increasing calcium concentration after a temperature jump (Fig. 3 A). $I_{ML1}$ is weaker following a temperature jump to 25°C at pCa 7 (Fig. 3 B) than in steady-state relaxing conditions (horizontal magenta line). A Hill fit of the $I_{ML1}$ data for pCa between 7 and 4.5 gave $pCa_{50}$ 6.56 ± 0.14 and $n_H$ 3.53 ± 1.83, showing that $I_{ML1}$ is much more sensitive to calcium than force development (Table 1). We also calculated the amplitude of the ML1 reflection ($A_{ML1}$), the square root of $I_{ML1}$, and fitted $A_{ML1}$ with the Hill equation (Fig. 3 C). $pCa_{50}$ for $A_{ML1}$ was 6.49 ± 0.10 and $n_H$ was 2.93 ± 1.94. Because the intensity of an x-ray reflection is to a good approximation proportional to the square of the number of contributing diffractors, we conclude that the fraction of myosin motors in the folded conformation, with a $pCa_{50}$ of 6.49, is much more sensitive to calcium than force development, with a $pCa_{50}$ of 5.99 (Table 1).

## Meridional x-ray intensity distribution

The meridian of the x-ray pattern in relaxing conditions at 25°C (Fig. 4, red) is dominated by a series of myosin-based reflections (M1 to M6) that are usually considered to be orders of the fundamental 43 nm periodicity of the myosin-based helix. However, the forbidden reflections (M1, M2, M4, and M5), so-called because they would not be present if the myosins were arranged in a perfect three-start helix, have additional components that index on a slightly longer 45.5 nm periodicity (Caremani et al., 2021). Both components are split into closely spaced subpeaks as a result of x-ray interference between the two motor arrays in each thick filament (Linari et al., 2000). Troponin-based reflections, indexing on an axial periodicity of ~38 nm, are also observed, with the first-order T1 reflection prominent.

All the myosin-based reflections apart from the M6, which is associated with a periodicity in the thick filament backbone, and the M3, which is associated with the periodicity of the myosin motors, are weaker following calcium activation. The M3 and M6 reflections remain strong but move to a slightly lower angle, signaling the characteristic increase in the axial periodicity of the myosin motors and thick filament backbone on activation. All these changes are similar to those observed during force development in an isometric tetanus in intact mouse EDL muscles at the same temperatures (Caremani et al., 2019).

## The M6 reflection

As noted above, the M6 reflection remains strong during active contraction (Fig. 4 and Fig. S2; Caremani et al., 2019). Its spacing ($S_{M6}$), which is thought to signal the axial periodicity of the thick filament backbone (Reconditi et al., 2004; Huxley et al., 2006), was higher following the temperature jump (Fig. 5) than during steady-state relaxation (horizontal magenta line), suggesting that the thick filament had not switched off fully after the temperature jump, in agreement with the behavior of the ML1 reflection (Fig. 3). At full calcium activation, $S_{M6}$ was about 7.28 nm at 25°C, the same as its value at the plateau of isometric

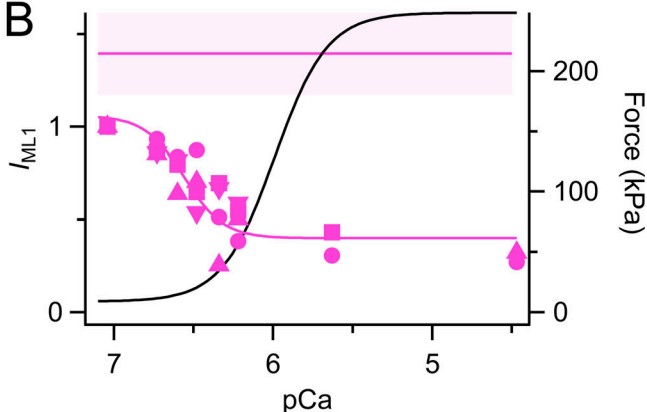

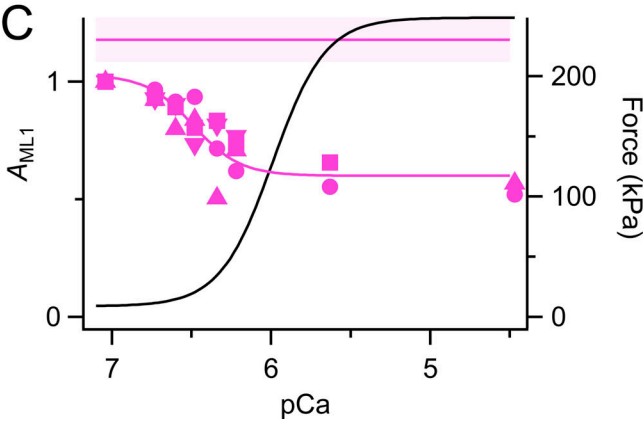

Figure 3. **First myosin layer line. (A)** Intensity distributions parallel to the meridional axis at different calcium concentrations (average of two to four bundles). pCa 9, red; pCa 7, orange; pCa 6.5, green; pCa 6.2, blue; pCa 4.5, violet. **(B)** Intensity of the first myosin layer line ($I_{ML1}$) was normalized by the value at pCa 7, following a temperature jump to 25°C. **(C)** Square root of $I_{ML1}$ ($A_{ML1}$). Magenta symbols and lines as in Fig. 2. Black lines are the Hill fit to force from Fig. 1 D.

tetanus in mouse EDL muscle at the same temperature (Caremani et al., 2019). The Hill fit to $S_{M6}$ had a calcium sensitivity ($pCa_{50}$) slightly greater than that for force (Table 1).

## The M3 reflection

The M3 reflection is associated with the axial repeat of the myosin motors along the thick filaments. It is strong in both

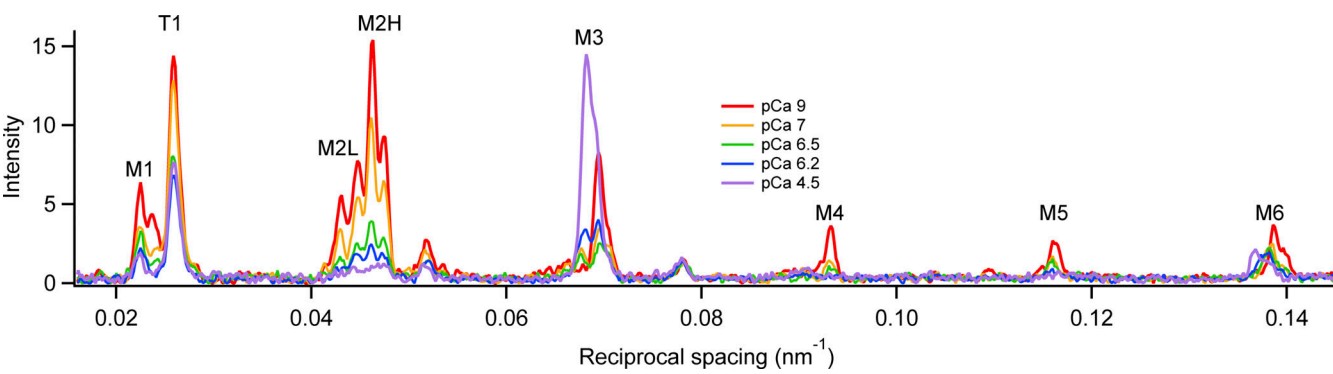

**Figure 4.** **Effect of calcium concentration on the meridional profiles.** Meridional intensity distributions at 25°C (average of two to four bundles) in relaxing solution (pCa 9, red) and after a temperature jump to 25°C pCa 7 (orange), pCa 6.5 (green), pCa 6.2 (blue), and saturating calcium concentration (pCa 4.5, violet).

relaxing and activating conditions, but its axial profile changes characteristically when calcium concentration is increased (Fig. 6 A), indicating movement of the average center of mass of the motors away from the midpoint of the myosin filament (Reconditi et al., 2011; Hill et al., 2021). The spacing of the reflection also increases, indicating an increase in axial periodicity proportional to that of the filament backbone, and the same as that observed at maximal activation at the same temperature in intact muscle (Caremani et al., 2019). The intensity of the M3 reflection ($I_{M3}$) increased monotonically with increasing calcium concentration after the temperature jump (Fig. 6 B). $pCa_{50}$ for the Hill fit to $I_{M3}$ was 6.07 ± 0.09 (Table 1), not significantly different from that for force. $pCa_{50}$ for $A_{M3}$, which is expected to be approximately proportional to the number of diffracting molecules contributing to the M3 reflection, was similar, 6.13 ± 0.07 (Table 1). Since myosin motors bound to actin with their long axes roughly perpendicular to the filament axis are likely to be the main contributors to the M3 reflection at high levels of activation (Reconditi et al., 2011), this result suggests that the

number of such motors is proportional to force in the steady state.

The spacing of the M3 reflection ($S_{M3}$), associated with the axial periodicity of the myosin motors, was consistently lower after a temperature jump ($S_{M3}$; Fig. 6 C) in relaxing conditions than in steady-state relaxation at the same temperature (horizontal magenta line). The incompletely off structure of the thick filament following a temperature jump is therefore associated with a slightly shorter axial periodicity of the motors, in contrast with the longer periodicity observed at higher levels of activation, and the longer periodicity of the myosin filament backbone reported in both conditions by $S_{M6}$ (Fig. 5). A similar decrease in $S_{M3}$ has been observed transiently during rapid shortening imposed soon after electrical stimulation in intact muscle (Brunello et al., 2006; Hill et al., 2022). $pCa_{50}$ for $S_{M3}$ was 6.24 ± 0.02, similar to that of $S_{M6}$, and slightly higher than that for force (Table 1).

An additional peak appears on the low-angle side of the M3 reflection during submaximal activation after a temperature jump (Fig. 6 A). This "star" peak corresponds to that reported previously both in intact mouse EDL muscle at rest and during steady-state relaxation of demembranated fibers from rabbit psoas muscle at pCa 9 at low temperature (Caremani et al., 2019, 2021), and during rapid shortening imposed soon after electrical stimulation of intact mouse EDL muscle at 28°C (Hill et al., 2022). It has a spacing between 14.75 and 14.85 nm (Fig. S3), which is too long for it to be considered as an interference subpeak of the M3 reflection. It is visible across the pCa range 7.0 to 6.2 following a temperature jump to 25°C (Fig. 6 A) and is almost as strong as the M3 reflection in that range. It was not detectable at high levels of calcium activation, but this may be due to the relatively high intensity of the nearby low-angle peak of the M3 reflection (Fig. 6 A, *la* peak) in those conditions.

The three interference subpeaks of the M3 reflection, labeled *la* (low angle), *ma* (mid angle), and *ha* (high angle) in Fig. 6 A, have been used extensively in previous studies to characterize axial displacements of the myosin motors (Piazzesi et al., 2002, 2007; Reconditi et al., 2011; Hill et al., 2021). The *la* peak, which is diagnostic for the presence of the roughly perpendicular actin-bound motors seen at the plateau of a tetanus in intact muscle (Reconditi et al., 2011; Caremani et al., 2019; Hill et al., 2021) was

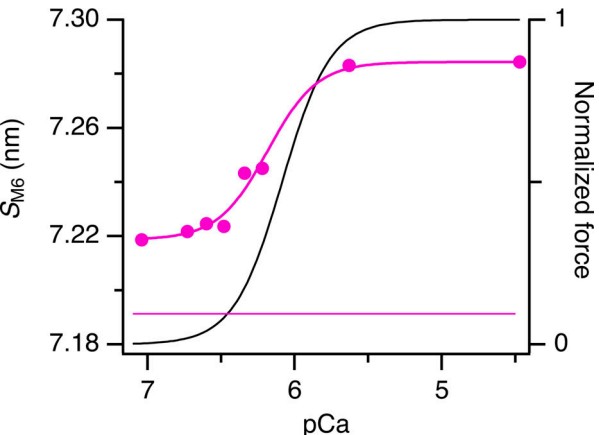

**Figure 5. Spacing of M6 reflection ($S_{M6}$) following a temperature jump to 25°C.** Meridional intensity profiles were averaged from $n$ = 4 bundles, except points at pCa 5.6 and 4.5, $n$ = 2 bundles. Hill equation fit is shown as a magenta line. Horizontal magenta line denotes steady-state value at pCa 9. Black line is the Hill fit to force from Fig. 1 D.

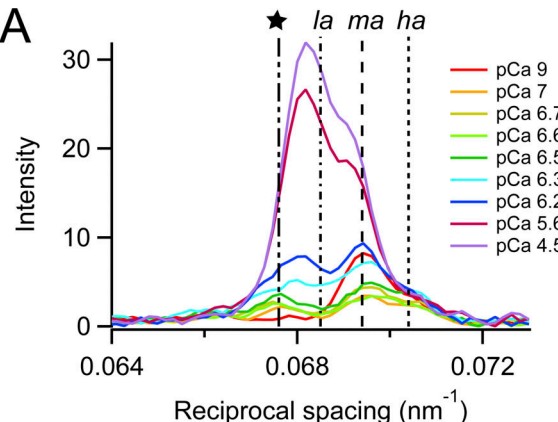

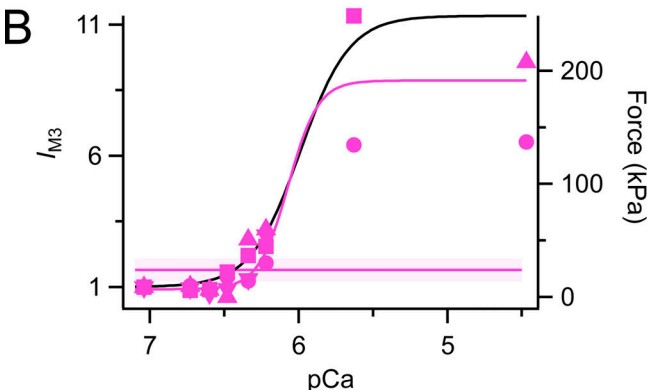

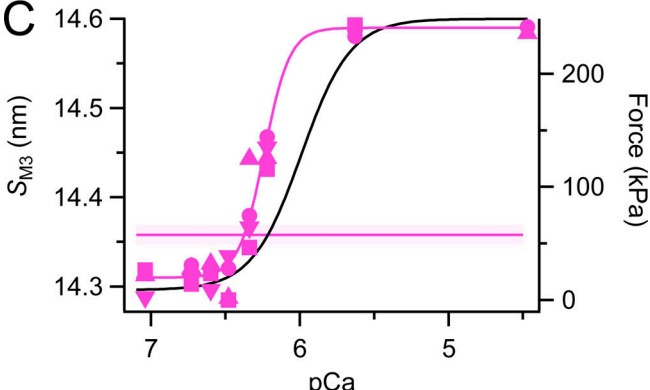

**Figure 6. M3 reflection. (A)** Meridional intensity distributions corrected by the cross-meridional width (average of two to four bundles). Red, pCa 9; orange, pCa 7; yellow, pCa 6.7; light green, pCa 6.6; dark green, pCa 6.5; cyan, pCa 6.3; blue, pCa 6.2; Bordeaux, pCa 5.6; and violet, pCa 4.5. Vertical dashed lines mark reciprocal spacings 1/(14.8 nm), 1/(14.6 nm), 1/(14.4 nm), and 1/(14.2 nm). **(B and C)** Intensity ($I_{M3}$, B) was corrected by its cross-meridional width and normalized by the value at pCa 7, and spacing ($S_{M3}$, C) following a temperature jump to 25°C. Magenta symbols and lines as in Fig. 2. Black lines are the Hill fit to force from Fig. 1 D.

not detectable up to pCa 6.4 (Fig. 6 A and Fig. S3). The relative intensity of the *ma* and *ha* peaks was also constant in these conditions, indicating that there is no change in the average axial displacement of the myosin motors at these low levels of calcium activation, despite the fact that the number of folded motors is decreasing in this range, as shown by $I_{ML1}$ (Fig. 3). At

lower pCa, the fractional intensity of the la peak, $L_{M3}$ (=$I_{la\ peak}$/$I_{M3}$) increased quite steeply so that $pCa_{50}$ for $L_{M3}$ was 6.22 ± 0.03 (Table 1), significantly higher than that for force.

**The M1 and M2 reflections**
The forbidden M1 reflection is considerably weaker following a temperature jump to 25°C at pCa 7 and below (Fig. 7 A) than in steady-state relaxation at pCa 9 (horizontal magenta line), and weakens further at increasing [Ca²⁺], though still detectable at full calcium activation. Although the M1 reflection has two or more subpeaks or components (Caremani et al., 2021), these were not clearly resolved in the present measurements. The relationship between the total intensity of the reflection ($I_{M1}$) and pCa was fitted by the Hill equation with $pCa_{50}$ 6.55 ± 0.06, much higher than that for force (Table 1).

Similar results were obtained for the M2 reflection, which however could be resolved into four subpeaks in relaxing conditions (Fig. 4), corresponding to pairs of interference subpeaks of the second orders of a fundamental L (lower angle) axial periodicity 45.5 nm and an H (higher angle) periodicity 43.1 nm corresponding to that of the myosin helix (Caremani et al., 2021). The L and H components could be resolved over the full range of pCa, allowing us to determine the pCa dependence of $I_{M2H}$ (Fig. 7 B) separately from that of $I_{M2L}$ (Fig. 7 C). Both sets of data were well fit by the Hill equation, with $pCa_{50}$ values 6.65 ± 0.06 and 6.63 ± 0.09, respectively, at 25°C, much higher than that of force (Table 1). $I_{M2H}$ and $I_{M2L}$ had already reached the low value, characteristic of full calcium activation at about pCa 6.4, when force is only about 15% of its maximum value. Hill fits of the amplitudes of the M2H and M2L reflections, $A_{M2H}$ and $A_{M2L}$, gave $pCa_{50}$ values indicating slightly lower calcium sensitivity, but still much higher than that of force (Table 1).

**The T1 reflection**
The troponin-based T1 reflection is weaker at full calcium activation than in relaxed or resting conditions (Matsuo et al., 2010), and this is likely to be due to myosin heads binding to preferred actin target sites midway between adjacent troponin core domains that are expected to dominate its periodic mass distribution along the thin filament. The intensity of the T1 reflection ($I_{T1}$), in contrast with most of the myosin-based reflections, is relatively insensitive to a temperature jump to 25°C in relaxing conditions (Fig. 8). $I_{T1}$ is however sensitive to increases in [Ca²⁺] that are subthreshold for force generation; the Hill equation fit to $I_{T1}$ at 25°C gave $pCa_{50}$ 6.61 ± 0.08, with a similar value for $A_{T1}$ (Table 1). This value is also similar to that for $I_{M2H}$ and $I_{M2L}$ and much higher than $pCa_{50}$ for force. $n_H$ for $I_{T1}$ and $A_{T1}$ is much larger than that for force.

## Discussion
### Slow recovery of the folded helical state of myosin following a temperature jump
X-ray diffraction patterns from bundles of muscle fibers contracting at different steady values of [Ca²⁺] were obtained by equilibrating them in a solution with the required [Ca²⁺] at 1°C and then rapidly raising the temperature to 25°C. This

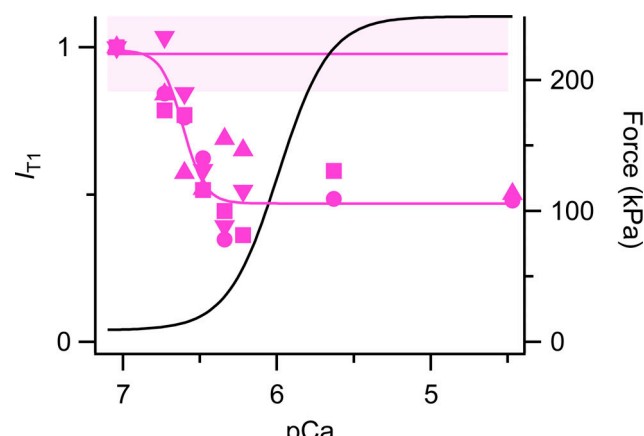

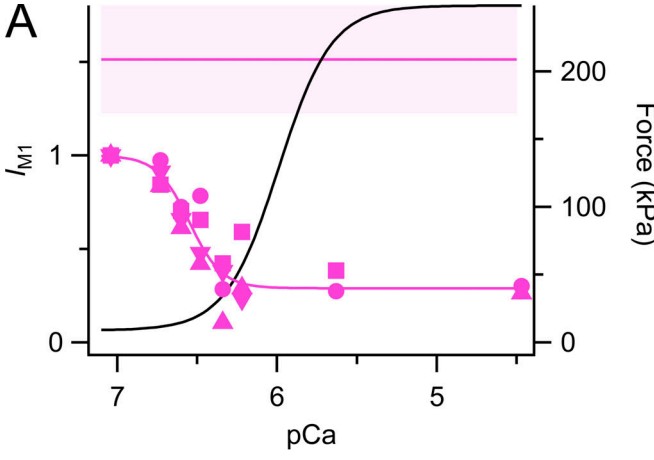

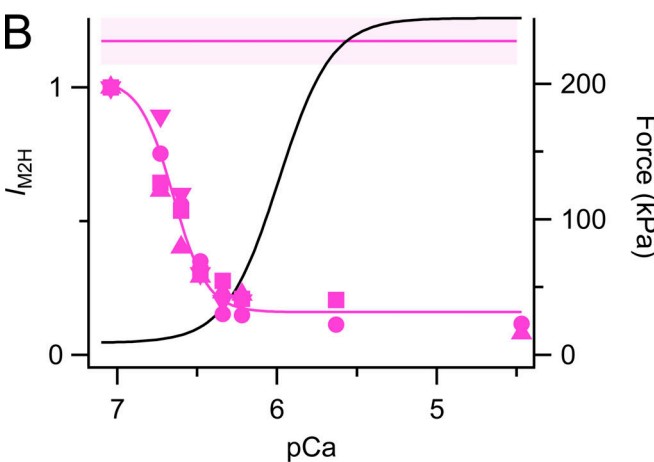

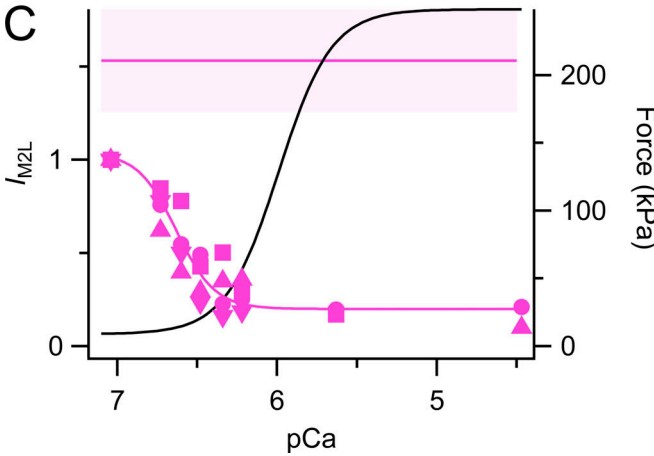

Figure 8. **Intensity of T1 reflection ($I_{T1}$) following a temperature jump to 25°C and normalized by the value at pCa 7.** Magenta symbols and lines as in Fig. 2. Black line is the Hill fit to force from Fig. 1 D.

first myosin x-ray layer line (ML1) had not completely recovered on the timescale of our experiments following the temperature jump to 25°C in relaxing conditions (Fig. 3). The same effect was present in other x-ray reflections that are sensitive to the OFF state (Figs 2, 5, 6, and 7). We further investigated this phenomenon in single demembranated muscle fibers using fluorescent probes on the myosin RLC that are sensitive to the same structural transition as the ML1 x-ray layer line (Fusi et al., 2015; Caremani et al., 2021). The results showed that, although there is a very fast (~100/s) component in the recovery of the OFF state following a temperature jump in relaxing conditions, as reported previously (Bershitsky et al., 2010), there is also a smaller slow component with a time constant of several seconds (Fig. 1, B and C). The incomplete recovery of the OFF state after a temperature jump observed in the x-ray experiments is likely to be due to this slow component of the refolding of the myosin motors.

The experiments on single muscle fibers also allowed temperature-jump activation to be compared with steady-state activation at a constant temperature of 25°C achieved using a conventional "pre-activation" in a relaxing solution containing a low concentration of calcium buffer (Fig. 1 B). The results showed that the calcium dependence of force in single muscle fibers expressed in terms of the Hill parameters $pCa_{50}$ and $n_H$ was similar after temperature jump activation and in steady-state activation using the conventional protocol (Fig. 1 A). Similar values of both parameters were reported recently for activation by temperature jump to 25°C in the same muscle fiber preparation, but in the absence of the RLC probes (Caremani et al., 2022). Thus, the incomplete recovery of the structural OFF state of the myosin motors soon after the temperature jump does not affect the steady-state force–pCa relationship. It also has no effect on $pCa_{50}$ and $n_H$ for the structural change in the myosin motors monitored by the RLC probes (Fig. 1 B), although it does reduce the total amplitude of the structural change between relaxation and full calcium activation by an amount corresponding to the slow component following the temperature jump.

There are additional effects associated with the use of fiber bundles in the temperature jump protocol to record the weaker

Figure 7. **Forbidden reflections. (A)** Intensity of M1 reflection ($I_{M1}$). **(B)** Intensity of M2H reflection ($I_{M2H}$). **(C)** Intensity of M2L reflection ($I_{M2L}$), all following a temperature jump to 25°C and normalized by the value at pCa 7. Magenta symbols and lines as in Fig. 2. Black lines are the Hill fit to force from Fig. 1 D.

temperature-jump activation protocol eliminates delays due to slow calcium ion diffusion into the bundles and minimizes irreversible structural changes associated with prolonged activation at 25°C (Linari et al., 2007). However, we found that the helical folded OFF state of the myosin motors signaled by the

x-ray reflections. $pCa_{50}$ and $n_H$ of force are much lower in the fiber bundles than in single fibers, which may be related to the greater end compliance of the fiber bundles, allowing greater sarcomere shortening during activation, and the likely build-up of inorganic phosphate in the bundles during force development after they have been removed from the bathing solution.

### The conformation of the myosin motors and the structure of the myosin filament at partial steady activation

The conformation of the myosin motors in actively contracting muscle has been characterized in many previous studies using the intensity and interference fine structure of the M3 reflection associated with the fundamental axial periodicity of the motors (Irving et al, 2000; Piazzesi et al, 2002, 2007, Reconditi et al, 2004, 2011; Hill et al, 2021). In the present experiments, the intensity of the M3 reflection ($I_{M3}$) had a relatively simple dependence on $[Ca^{2+}]$, increasing in parallel with active force at increasing levels of steady calcium activation (Fig. 6 B). This relationship is consistent with the progressive replacement of the folded OFF conformation of the motors with the actin-attached force-generating conformation associated with maximal isometric contraction, in which the long axis of the motors is roughly perpendicular to the filament axis, and the fraction of motors in that conformation is directly proportional to the active force (Brunello et al., 2006).

In the previous studies mentioned above, further information about the conformation of the myosin motors was obtained from the interference fine structure of the M3 reflection, which gives a precise measure of their average axial center of mass. In the present experiments, this approach was complicated by the presence of a new reflection on the low-angle side of the M3, corresponding to a periodicity of about 14.8 nm, called the star peak (Fig. 6 A). The star peak is not detectable in steady-state relaxing conditions at pCa 9, but at low levels of partial calcium activation following a temperature jump, when the thick filament is partly ON, the star peak has about the same intensity as the M3 itself (Fig. 6 E). Moreover, the spacing of the M3 reflection, $S_{M3}$, takes up the anomalous value of 14.30 nm, lower than that during steady state relaxation at pCa 9, and no longer simply related to the periodicity of the filament backbone reported by $S_{M6}$, which increases during partial activation (Fig. 5). Analysis of the interference fine structure of the M3 reflection (Fig. S3) showed that its $la$ component, which signals the presence of the perpendicular actin-attached force-generating myosin motors characteristic of full calcium activation, was undetectable in the pCa range 7 to 6.4, and the relative intensities and spacings of the $ma$ and $ha$ components were independent of pCa in this range. As activating $[Ca^{2+}]$ is increased in the range pCa 7 to 6.4, the fraction of motors in the helical folded conformation decreases by about 30% based on the change in $A_{ML1}$ (Fig. 3 C), with little change in the intensity (Fig. 6 B) or amplitude (Fig. S3 A) of the M3 reflection. The folded helical motors that are lost as $[Ca^{2+}]$ is increased in this range are not replaced by motors in the conformation characteristic of full calcium activation, but rather by a distinct motor conformation with a longer axial periodicity, about 14.8 nm, called the star peak in Fig. 6.

The origin of the star peak is unknown. Its spacing is intermediate between the expected values for the third orders of the two axial periodicities observed in relaxed muscle (Caremani et al., 2021), and seen here as the L and H components of the M2 reflection (Fig. 7). The H periodicity indexes on 43.1 nm, exactly matching the helical periodicity of ML1, but the structural basis of the L periodicity, which is also seen in heart muscle (Ovejero et al., 2022), is unknown. The interference fine structure of the L components of the M2 reflection suggested that the diffracting structures are confined to the C zone of the filament, both in skeletal and heart muscles (Caremani et al., 2021; Ovejero et al., 2022), and on that basis, we previously suggested that the L periodicity might be related to the super-repeat organization of C-type titin domains in the filament backbone. However, that hypothesis is inconsistent with recent high-resolution cryo-EM structures of the myosin-binding protein-C-containing C zone of cardiac thick filaments in the presence of the myosin inhibitor mavacamten, either isolated (Dutta et al., 2023 Preprint) or in the myofibrillar lattice (Tamborrini et al., 2023 Preprint), which showed only a single axial periodicity matching that of the helix of myosin motor domains.

There are currently no high-resolution structures of the D (distal to the C zone) zone of the thick filament, which has a distinct titin super-repeat, so it remains possible that the longer myosin-based L periodicity seen by x-ray diffraction is associated with a distinct packing of myosin tails and titin in the D zone. The relatively high intensity of the L-periodicity reflections in relaxed muscle, and of the star peak compared with that of the main M3 reflection in the pCa range 7 to 6.4, make it highly likely that they are associated with myosin motors; no other thick filament component has the appropriate periodic mass distribution. The fact that only one axial periodicity is observed in cryo-EM structures of the C zone of thick filaments in the presence of mavacamten suggests that the L motors are in the D zone. The star reflection, which has a periodicity between that of the third order of the fundamental L and H periodicities, could then be the result of x-ray interference between L motors in the D zone and H motors in the C zone.

The star peak has not been detected in the fully OFF state of either heart or skeletal muscle, either intact or demembranated, but is not simply a peculiarity of the partially activated state produced by a temperature jump to 25°C. It has also been observed on cooling of either intact or demembranated muscle (Caremani et al., 2019, 2021; Ovejero et al., 2022). Moreover, it appears transiently during force development in the tetanic response to repetitive electrical stimulation in intact EDL muscle of mouse at 28°C, when force is about 20% of its maximum value, although it cannot be detected at the tetanus plateau (Hill et al., 2021). The star peak was also detected at the peak of the twitch response to a single stimulus. When rapid unloaded shortening was imposed at the start of tetanus, delaying the formation of strong actin-bound motors and force development, the star peak was visible for a longer period at forces between 0 and about 50% of maximum. At the same time, $S_{M3}$ decreased to 14.31 nm whilst $S_{M6}$ increased slightly, and the intensity of the $la$ subpeak of the M3 reflection decreased slightly (Hill et al., 2022). The intensity of the ML1 reflection signaling the helical

folded state was already greatly reduced when the star peak was observed.

All these features are characteristic of the partially active state described here. Although the time courses of the M1, M2, and T1 reflections have not been reported in the same preparation and conditions, it is clear from previous x-ray experiments on twitches and tetani in frog muscle fibers (Matsuo and Yagi, 2008; Matsuo et al., 2010; Reconditi et al., 2011) that the intensities of all these reflections are lost much faster than force development. Thus, although the star peak was not clearly resolved during the activation of single fibers from frog muscle (Reconditi et al., 2011), the available evidence is consistent with the intermediate activation state described here being formed transiently during the physiological activation of mammalian muscle.

### Implications of the distinct calcium sensitivities of different x-ray reflections for potential mechanisms of thick filament activation

The calcium sensitivity of force and of the x-ray and probe signals associated with different aspects of thick filament structure was described in terms of the Hill equation parameters $pCa_{50}$ and $n_H$ that gave the best fit to each set of data in the pCa range 7–4.5 (Table 1). In the conditions of the present experiments, $pCa_{50}$ for force was $5.99 \pm 0.17$, which is not significantly different from that for $A_{M3}$, $6.13 \pm 0.07$. $A_{M3}$ is expected to track the number of force-generating motors strongly bound to actin with their long axes roughly perpendicular to the filament axis. A second set of x-ray signals, including the equatorial intensity ratio $I_{11}/I_{10}$ and the axial periodicities of the filament backbone and myosin motors, $S_{M6}$ and $S_{M3}$ respectively, have slightly higher calcium sensitivities. $I_{11}/I_{10}$ signals the movement of myosin motors towards the thin filaments, and its slightly greater calcium sensitivity may reflect sarcomere shortening at low levels of activation, as implied by the increase in $d_{10}$ (Fig. 2 C), which is expected to increase $I_{11}/I_{10}$. The longer thick filament periodicities $S_{M6}$ and $S_{M3}$ are associated with mechanosensing (Linari et al., 2015), and here the higher calcium sensitivity may be a consequence of the non-linear dependence of filament periodicity on force (Linari et al., 2015; Piazzesi et al., 2018).

Much higher calcium sensitivities were observed for the intensities and amplitudes of the forbidden x-ray reflections M1 and M2 that signal regular perturbations in the myosin helix in the relaxed state (Huxley and Brown, 1967), for which $pCa_{50}$ was 0.5–0.7 pCa units larger than that for force. $n_H$ was also higher for these x-ray reflections, suggesting that the loss of the helical perturbations is a cooperative process, as might be expected for a structural transition involving multiple intermolecular interactions between myosin motors. The intensity and amplitude of the ML1 layer line have an intermediate calcium sensitivity but still greater than that for force by about 0.5 pCa units. At the calcium concentration at which the intensities of the M2H and M2L reflections are reduced to about half of their relaxed values, the force is only about 5% of the maximum. This difference could only be explained on the basis of the mechanosensing hypothesis of thick filament activation if the relationship between filament

activation and stress were extremely non-linear. In addition, the distinct calcium sensitivities of the different x-ray reflections and the appearance of the star peak in the pCa range 7 to 6.4 imply that the structure of the thick filament at partial activation is not a simple mixture of OFF and ON states, which might also imply the presence of multiple thick filament activation mechanisms.

We consider two general mechanisms of thick filament activation that could operate in parallel with mechanosensing: direct calcium binding to a putative regulatory site on the thick filament or on titin and thin-to-thick filament signaling. The orientation of bifunctional probes on the myosin RLC in demembranated fibers from rabbit psoas muscle is not sensitive to calcium when force is inhibited by blebbistatin (Fusi et al., 2016), but it might be argued that stabilization of the myosin filament OFF state by blebbistatin could block a direct calcium response of the thick filaments. Electrical stimulation of intact fibers from frog skeletal muscle in which active force was inhibited by para-nitro-blebbistatin produced a large increase in the stiffness of I-band titin (Squarci et al., 2023), likely mediated by a calcium-dependent interaction of I-band titin with the thin filament. This could provide a mechanism of signaling to the thick filament through A-band titin, and a stretch of the fibers in these conditions revealed a hierarchical sequence of changes in thick filament structure and myosin motor conformation associated with filament activation.

Evidence in support of a direct effect of calcium on the myosin filaments in heart muscle came from studies on permeabilized pig heart in which force was abolished by a small molecule (MYK-7660) that is thought to inhibit thin-filament activation (Ma et al., 2022). Those studies showed that some x-ray signals associated with thick-filament activation in steady-state calcium titrations were still present at saturating concentrations of MYK-7660, although in some cases with reduced amplitude. However, there are several striking differences between the calcium dependence of the x-ray signals in that study and those reported here. For example, increasing $[Ca^{2+}]$ in the range pCa 9–4.5 greatly reduced the M3 and M6 reflections in that study, even in control conditions, whereas here the intensity of the M3 reflection increased dramatically with increasing $[Ca^{2+}]$ (Fig. 6 B), as expected from the formation of perpendicular actin-bound motors, and the intensity of the M6 reflection was almost independent of $[Ca^{2+}]$ (Fig. 4), as expected from its proposed origin in the thick filament backbone. The unexplained differences between the results of the two studies make extrapolation of the interpretations unreliable. The present results do not exclude the possibility of direct calcium binding to the thick filament but neither do they provide definitive evidence for it.

The thin-to-thick filament signaling hypothesis would postulate that calcium does not bind directly to the thick filament but activation of the thin filament by calcium is somehow sensed by the thick filament, possibly via titin (Squarci et al, 2023). The potential role of this mechanism in the present experiments is difficult to assess in the absence of a reliable measure of the activation state of the thin filament as a function of $[Ca^{2+}]$. The intensity of the troponin reflection ($I_{T1}$), which is associated with

the axial repeat of troponin in the thin filaments, might be considered such a measure and has the same high calcium sensitivity and cooperativity ($n_H$) as $I_{M2L}$ and $I_{M2H}$ (Fig. 8 and Table 1). However, the interpretation of changes in $I_{T1}$ is uncertain because it is sensitive to the binding of myosin motors to actin target zones between troponin core domains as well as to conformational changes in troponin related to thin filament activation (Matsuo and Yagi, 2008; Sugimoto et al., 2008; Matsuo et al., 2010; Reconditi et al., 2011).

The orientation of bifunctional fluorescent probes bound to the calcium-binding subunit of troponin, TnC, was recently measured in calcium titrations in the same preparation and experimental conditions (temperature and dextran concentration) as the present experiments (Brunello et al., 2023). $pCa_{50}$ for TnC probe orientations in those calcium titrations were slightly smaller than those for force, and there was no evidence for a high calcium-sensitivity component of the orientation change in the range of $[Ca^{2+}]$ in which we observed changes in $I_{T1}$, $I_{M2L}$, and $I_{M2H}$. We conclude that it is unlikely that these x-ray signals are part of a structural pathway in which the canonical activation of the thin filament by calcium binding to TnC is transmitted to the thick filament; they seem more consistent with the presence of an additional mechanism of thick filament activation that is more calcium sensitive than TnC.

In summary, the present results suggest that thick filament activation in the steady state, and probably also in response to the physiological calcium transient, involves multiple mechanisms and intermediate structural states with distinct calcium sensitivities. The present results start to reveal the structural basis of these additional mechanisms, but fundamental questions remain about the potential role of direct calcium binding to the thick filaments and interfilament signaling. X-ray diffraction and interference from demembranated muscle cells in situ, combined with improved annotation of the x-ray reflections from cryo-EM structures of the filaments in defined static states, will provide a powerful approach to answering these questions in future studies.

### Data availability
All relevant data, associated protocols, and materials are within the manuscript and its online supplemental material. If any additional information is needed, it will be available upon request from the corresponding author.

## Acknowledgments
Henk L. Granzier served as editor.

We thank the ESRF for the provision of synchrotron beamtime; Mario Dolfi, the staff of the mechanical workshop of the Department of Physics and Astronomy (University of Florence), and Jacques Gorini (ESRF) for electronic and mechanical engineering support.

This work and some investigators were supported by Fondo per gli Investimenti della Ricerca di Base (Futuro in Ricerca project RBFR08JAMZ to E. Brunello, Italy); Italian Ministry of Education, University and Research (MiUR-PRIN 2010R8JK2X_006 and 20207P85 MH); Italian Telethon Foundation (grant number GGP12282); Consorzio Nazionale Interuniversitario per le Scienze Fisiche della Materia (Progetto d'Innesco della Ricerca Esplorativa 2007; Italy); Fondazione Cassa di Risparmio di Firenze (2016–2018; Italy); UK Medical Research Council (grant MR/M026655/1); and the ESRF. M. Caremani was funded by the University of Florence (competitive projectmarcocaremani_rictd1819) (Italy). L. Fusi was funded by a Sir Henry Dale Fellowship awarded by the Wellcome Trust and the Royal Society (Fellowship 210464/Z/18/Z). E. Brunello was funded by a British Heart Foundation Intermediate Basic Science Research Fellowship FS/17/3/32604. This research was funded in part by the Wellcome Trust (Grant: 210464/Z/18/Z). For the purpose of open access, the author has applied a CC BY public copyright license to any author-accepted manuscript version arising from this submission.

Author contributions: M. Caremani, L. Fusi, M. Reconditi, G. Piazzesi, T. Narayanan, M. Irving, V. Lombardi, M. Linari, and E. Brunello designed and performed research; L. Fusi and E. Brunello analyzed data; M. Caremani, L. Fusi, M. Linari, M. Reconditi, G. Piazzesi, T. Narayanan, M. Irving, V. Lombardi, and E. Brunello wrote and contributed to critical revision of the paper.

Disclosures: The authors declare no competing interests exist.

Submitted: 24 March 2023

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

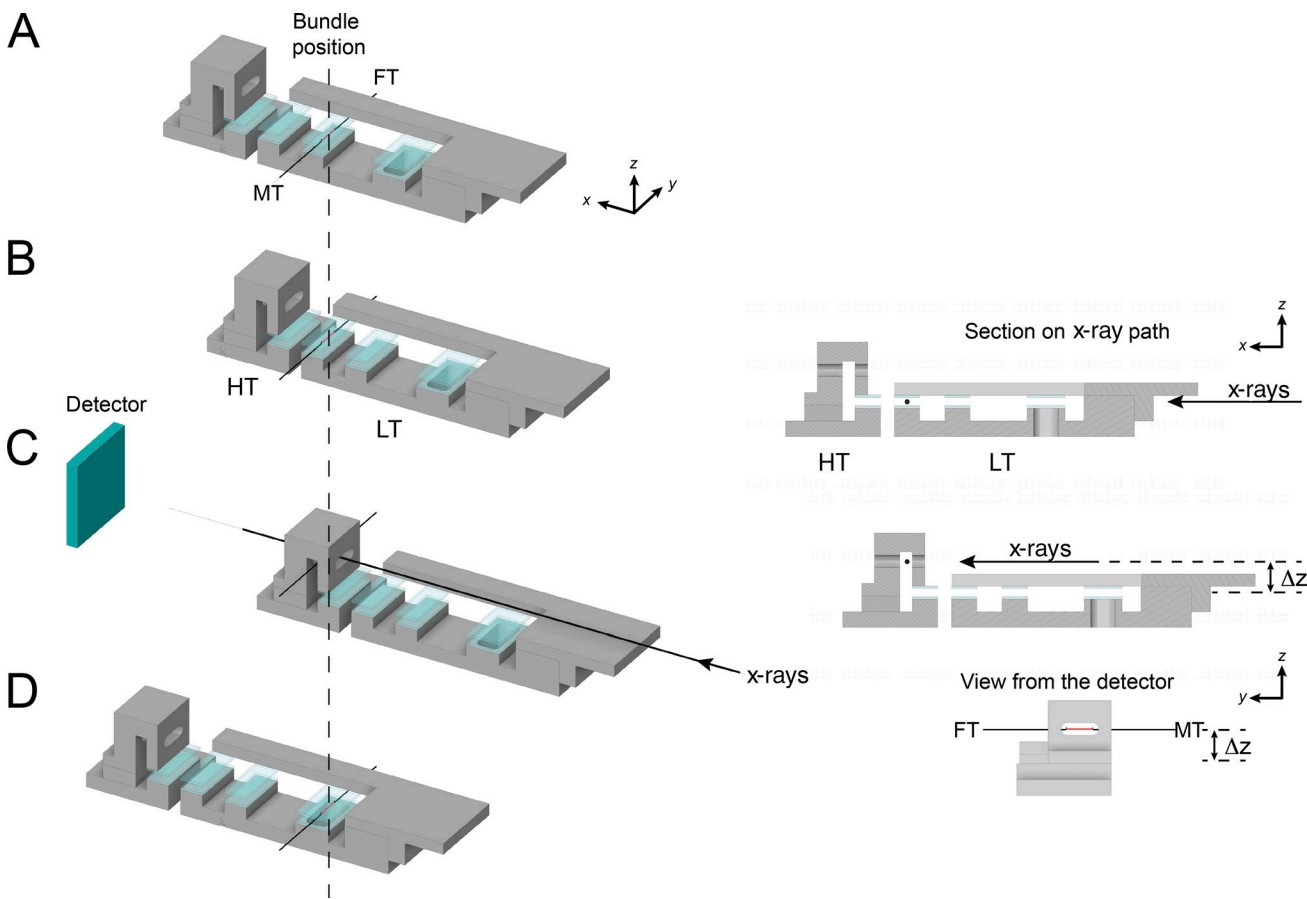

Figure S1.   **Schematic of the x-ray experimental setup.** The bundles were horizontally mounted between the levers of a force transducer (FT) and a motor (MT) in a thermoregulated multidrop mechanical apparatus that allowed activation using a temperature jump technique (Linari et al., 2007). The positions of the bundle (vertical dashed line) and of the x-rays path were fixed, while the trough could move horizontally and vertically (x and z axis, respectively). **(A–C)** The bundle (red) was kept in a preactivating solution at low temperature (1°C, LT) for 2 min (A), then transferred to the activating solution at 1°C by horizontal shift of the trough (B), in which little force was developed. When force became steady (within 10 s), the bundle was transferred to an activating solution at 25°C (HT) and, following full force development, the trough was lowered (Δz) so that the bundle was transferred to air in a narrow aluminum-delimited chamber for the x-ray exposure (C, and xz sections and view along the x axis from the detector on the right). Exposure in air avoids x-ray absorption by the solution drops. **(D)** The bundle was then relaxed in a relaxing solution. The detector in C would be at 5 m from the bundle but drawn closer in the schematic.

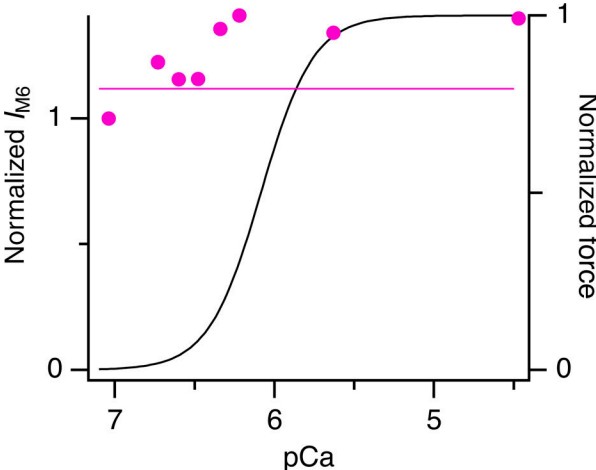

Figure S2.  **Intensity of M6 reflection ($I_{M6}$) following a temperature jump to 25°C and normalized by the value at pCa 7.** Data are corrected by the cross-meridional width. Due to the lower signal-to-noise, the meridional intensity distributions of the M6 reflection were averaged from two to four bundles before determining its intensity at each pCa. Horizontal magenta line denotes steady-state value at pCa 9. Black line is the Hill fit to force from Fig. 1 D.

Figure S3.  **M3 component peaks and star peak. (A)** Square root of the M3 intensity. **(B)** Fractional intensity of low-angle peak ($L_{M3} = I_{la\ peak}/I_{M3}$). **(C)** Intensity of star peak ($I_{star}$). **(D and E)** Spacing (D) and intensity (E) of the star (cyan) and M3 component peaks (green, *la*; purple, *ma*; orange, *ha*) following a temperature jump to 25°C. Different symbols identify individual fiber bundles. Horizontal lines and shading (mean ± SD, *n* = 4 bundles) denote steady-state values at pCa 9. Note that the *la* and star peak are not visible at pCa 9. Black lines are the Hill fit to force from Fig. 1 D.

