## [Peer Review File · The Journal of General Physiology]

Dependence of myosin filament structure on intracellular calcium concentration in skeletal muscle

Marco Caremani, Luca Fusi, Massimo Reconditi, Gabriella Piazzesi, Theyencheri Narayanan, Malcolm Irving, Vincenzo Lombardi, Marco Linari, and Elisabetta Brunello

Corresponding Author(s): Elisabetta Brunello, King's College London and Marco Linari, University of Florence

Review Timeline:

Submission Date:	March 24, 2023
Editorial Decision:	May 9, 2023
Revision Received:	July 15, 2023
Editorial Decision:	August 15, 2023
Revision Received:	August 31, 2023

Editor: Henk Granzier

Transaction Report:

DOI: <https://doi.org/10.1085/jgp.202313393>

May 9, 2023

Dr. Elisabetta Brunello
King's College London
Randall Centre for Cell and Molecular Biophysics
New Hunt's House, Guy's Campus
London SE1 1UL
United Kingdom

Re: 202313393

Dear Dr. Brunello,

Thank you for submitting your manuscript, entitled "Dependence of myosin filament structure on intracellular calcium concentration in skeletal muscle" to JGP. Your manuscript has now been seen by 3 reviewers, whose comments are appended below. You will see that the reviewers have raised several concerns that should be addressed prior to further consideration of the manuscript at JGP.

We would be pleased to receive a suitably revised manuscript that addresses these concerns, which will be re-reviewed, most likely by some or all of the original referees. Based on the scope of the requested changes, we typically anticipate that the revision process will take no longer than 6 months, however, we understand you may need additional time to work on your resubmission to JGP. We therefore ask that you simply keep us informed as to a realistic submission timeline that is appropriate for your particular circumstances. In addition, please do not hesitate to contact me (via the editorial office) if you feel that a discussion of the reviewers' and editors' comments would be helpful.

Please submit your revised manuscript via the link below along with a point-by-point letter that details your responses to the editors' and reviewers' comments, as well as a copy of the text with alterations highlighted (boldfaced or underlined). If the article is eventually accepted, it would include a 'revised date' as well as submitted and accepted dates. If we do not receive the revised manuscript within one year, we will regard the article as having been withdrawn. We would be willing to receive a revision of the manuscript at a later time, but the manuscript will then be treated as a new submission, with a new manuscript number.

Please pay particular attention to recent changes to our instructions to authors in sections: Data presentation, Blinding and randomization and Statistical analysis, under Materials and Methods, as shown here: <https://rupress.org/jgp/pages/submission-guidelines#prepare>. Re-review will be contingent on inclusion of the required information (including for data added during revision) and demonstration of the experimental reproducibility of the results (i.e., all experimental data verified in at least 2 independent experiments).

Please note, JGP now requires authors to submit Source Data used to generate figures containing gels and Western blots with all revised manuscripts (when applicable). This Source Data consists of fully uncropped and unprocessed images for each gel/blot displayed in the main and supplemental figures. If your paper includes cropped gel and/or blot images, please be sure to provide one Source Data file for each figure that contains gels and/or blots along with your revised manuscript files. File names for Source Data figures should be alphanumeric without any spaces or special characters (i.e., SourceDataF#, where F# refers to the associated main figure number or SourceDataFS# for those associated with Supplementary figures). The lanes of the gels/blots should be labeled as they are in the associated figure, the place where cropping was applied should be marked (with a box), and molecular weight/size standards should be labeled wherever possible. Source Data files will be made available to reviewers during evaluation of revised manuscripts and, if your paper is eventually published in JGP, the files will be directly linked to specific figures in the published article.

Source Data Figures should be provided as individual PDF files (one file per figure). Authors should endeavor to retain a minimum resolution of 300 dpi or pixels per inch. Please review our instructions for export from Photoshop, Illustrator, and PowerPoint here: <https://rupress.org/jgp/pages/submission-guidelines#revised>

When revising your manuscript, please be sure it is a double-spaced MS Word file and that it includes editable tables, if appropriate.

Please submit your revised manuscript via this link:
Link Not Available

Thank you for the opportunity to consider your manuscript.

Sincerely,

Henk L. Granzier, Ph.D.
On behalf of Journal of General Physiology

Journal of General Physiology's mission is to publish mechanistic and quantitative molecular and cellular physiology of the highest quality; to provide a best-in-class author experience; and to nurture future generations of independent researchers.

Reviewer #1 (Comments to the Authors):

General comments:

In this manuscript by Caremani et al., the authors present some provocative and interesting X-ray diffraction data from skinned skeletal muscle obtained using an uncommon approach, namely temperature jump as a means to rapidly activate skeletal muscle fibers at different pCa values as well as avoid excessive rundown at high temperatures. The methods and data presentation are very clear. The primary findings presented that X-ray signals associated with the perpendicular motors characteristic of isometric force generation had the same calcium sensitivity as force, but X-ray signals associated with perturbations in the folded myosin helix had a much higher calcium sensitivity and that there is a population of myosin motors with a longer axial periodicity than the resting periodicity is seen at low levels of calcium activation, and proposed as an intermediate regulatory state of the myosin motors in the physiological pathway of filament activation. While I have some methodological concerns, the presentation of these findings is clear.

There are a number of awkwardly written and confusing sections in this paper, mostly in the discussion but also in the introduction. For instance, in the abstract and introduction they set things up so that a goal of the paper appears to be to test the hypothesis that there is a direct effect of calcium on structural off to on transitions skeletal muscle, as previously demonstrated in cardiac muscle. Indeed, the title of the paper is "Dependence of myosin filament structure on intracellular calcium concentration in skeletal muscle." In the abstract they say "High-force contraction depends on the release of the folded motors, which can be triggered by stress in the thick filament backbone, but additional mechanisms may link the activation of the thick filaments to that of the thin filaments or to intracellular calcium concentration." In the discussion they argue that their data indicates a need for some other activation mechanism in addition to mechano-sensing but calcium activation per se is discussed only very obliquely in the discussion. Since this was proposed as a guiding hypothesis in the introduction it should be either discussed more directly or set things up differently so it does not appear to be a primary focus of the paper, which I would suggest that it isn't.

There are also concerns with the data themselves. The X-ray data are very lean (n=2 for higher calcium measurements) making it difficult to assess the statistical significance of the findings. The choice of 5% dextran to restore the in vivo lattice spacing is also problematic since the authors have shown in previous publications this significantly over-compresses the myofilament lattice to non-physiological lattice spacings (see below). Another concern is the extent to which the OFF-state deficit after a T-jump could be obscuring events that happen at low calcium under more physiological conditions.

Specific Concerns

Introduction

The mechanical protocols that were used to establish the role of mechano-sensing to activate thick filaments were confined to a few tens of milliseconds after stimulation, and do not exclude direct effects of calcium on slower time scales.

Moreover, recent evidence suggests that the thick filaments in heart muscle can be directly regulated by calcium (Ma et al., 2022).To test for such a mechanism in skeletal muscle, we determined the dependence of thick filament structure on intracellular free calcium concentration in the steady state.

But the authors don't actually test for a direct calcium effect on thick filament regulation in their experiments. Actin binding was not inhibited in this study so it would not be possible to observe calcium dependent thick filament activation in isolation. Setting it up this way is misleading. What they are observing is the effects of calcium on both thick and thin filament structure together. The best solution perhaps would be to simply discuss calcium dependent thick filament regulation as one of the possible other mechanisms that could supplement mechano-sensing when they discuss the need for such mechanisms.

Specific Concerns

Introduction.

No previous X-ray study, to our knowledge, has described the calcium dependence of myosin conformation and thick filament structure during active contraction at 25 deg. C in the presence of 5% Dextran.

They need to specify "skeletal muscle" in this sentence. High temperature and 3% dextran was looked at in cardiac muscle eg. Ma et al., 2022 JGP. (paper quoted in this MS).

As indicated above, there are good reasons for avoiding 5% dextran. Previous publications by the authors previously showed (Caramini et al. 2021) 5% is substantially more than required to restore in vivo lattice spacing (overshoots by ~2 nm). 3-4% are more commonly used concentrations This needs to be pointed out and its implications discussed.

Similarly, in the methods: The osmotic agent dextran T500 (5% w/v) (Pharmacia Biotech, Uppsala, Sweden) was added to the experimental solutions to reduce the interfilament spacing to a value similar to that of intact muscle (Linari et al., 2007; Caremani et al., 2019; Caremani et al., 2021)

This is potentially misleading. Depends what you mean by "similar" I don't think they are all that "similar"

Under these experimental conditions, the activation of the myosin motors on the thick filament in demembrated skeletal muscle fibres is controlled by a stress-dependent mechanism that does not depend on the calcium concentration when active force generation is inhibited by Blebbistatin (Fusi et al., 2016).

I don't think this observation really helps here. Blebbistatin has been showed to favor the closed state of myosin heads in solution studies (Gollapudi et al., 2021 PMID: 34627791) and sequester myosin heads in a helically ordered state that doesn't reverse quickly. It could be that Blebbistatin stabilized heads are just not sensitive to calcium but are sensitive to strain. Since Blebbistatin stabilized heads may not correspond to a physiological state, I would just omit this sentence here or use it as part of an argument in the discussion where other possible regulatory mechanisms are discussed (see below)

I think the introduction could have been usefully concluded with the main takeaways of the paper that were outlined in the abstract to keep the reader focused on the primary findings.

Results

Force/pCa done conventionally and with T-jump normalized to Fmax agree reasonably well. Given that the T-jump seems to result in large offsets in the structural measurements, at low calcium it would be desirable to compare the force/unit area tension values

The X-ray data in Figure 2-5 are pretty lean with only 4 biological replicates for low force points and 2 for the high. I appreciate that it would not be trivial to get more data but we need to be reassured that the conclusions are sound. For instance, in Figure 2 - N=2 for the high force points, so that fit to the I11/I10 curve is not very good. Error bars don't mean much for this low number of n's and nH and pCa50 values are probably not very accurate.

Several of the fits are to summed X-ray data (Fig 2,3, 5) so there is essentially one measurement per point. While the points in these figures relatively show low scatter (compared to figure 2), this makes it very hard to demonstrate significant differences in derived values in Hill fits. Not sure what to make of the error bars on the Hill fits if there are no error bars in the data. While I tend to believe the conclusions that many of the structural changes are more sensitive to calcium than to force, statistically this might be hard to support.

It would have been better to calculate Hill fits for individual preps (replicates), then take the values for all the preps and average them and do stats. This would be much more accurate and confidence inspiring. Obviously, this doesn't work for the summed data.

Discussion

It seems clear that you don't get full restoration of off state after a T-jump at pCa 7. This would seem to complicate interpretation of structural events during activation if a T-jump left heads in a partially ON state. Looking at figure 7 for example it appears that about 1/3 of layer line intensity has gone at the end of the T-jump. In Figure 5 Sm6 indicates that thick filaments about 0.4 % longer than before T-jump. The "structural deficit" in the layer lines and M6 implies a great deal of the initial stages of thick filament activation have already occurred at the end of the T-jump in the absence of calcium. If so, the T-jump might overshadow or possibly blunt activation processes that occur at submaximal calcium. The authors should discuss how does this affects their general conclusions, especially efforts to characterize the transient states indicated by the *state

"Implications for mechanosensing"

You would a priori expect a sequence of structural events during activation so it is not surprising to me that all the structural

events don't all track force. Many of the changes require force generating heads but you would expect the loss of helical ordering to start before significant amount of force is generated. This is in fact, what is seen here.

If myosin activation were directly proportional to force in the steady state, the calcium dependence of structural signals related to filament activation. would be expected to be the same as that of force.

I would expect this only for reflections due to myosin head bound to actin. SM3 increase and IM3 increase should indeed match force which it does.

In contrast, pCa50 for SM6 is much closer to that of force, and nH is the same as that for force within the precision of the measurements. pCa50 for IM3 and AM3 are also close to that of force.

Again this is not unexpected.

"This is clearly not the case (Table 1). pCa50 for IML1 or AML1 is larger, by almost 0.3 pCa units, than that for force, as is that for the orientation of the RLC region of the myosin motors (Fig. 1B), which also has a higher value of the Hill coefficient nH."

And

"pCa50 for the intensities or amplitudes of the forbidden X-ray reflections M1 and M2 that signal regular perturbations in the myosin helix in the relaxed state (Huxley and Brown, 1967) is 0.4-0.5 pCa units larger than that for force, and nH is again higher."

and

Logically, the loss of helical order would be expected to precede binding so it should also precede force, which the authors show that it does. This is what you expect if there was a calcium -dependent a thick filament-based activation process as shown in cardiac muscle. There may be other possible explanations for this other than direct calcium effects on thick filament structure, but these possibilities need to be discussed. In any case, it seems an odd omission not to discuss what was appeared to be proposed in the introduction as a hypothesis they were testing. The only thing said appears to be:

"It is also possible that the myosin-based structural changes are controlled by a signaling pathway independent of the regulatory calcium sites on troponin. The resolution of those possibilities will require further studies."

Hard to argue with but does not directly address the question.

In this context, the discussion of a non-linear response of activation to force seems a bit forced and there are simpler mechanisms that can be imagined, providing that one accepts that things other than force might be involved in thick filament activation

Implications for the physiological regulation of muscle contraction

The above-mentioned concerns about how non-physiological aspects of the T-jump and the high degree of osmotic compression may be obscuring things should be discussed.

General comment:

It is disappointing that there isn't more reference to work by other investigators who have done relevant work.

Reviewer #2 (Comments to the Authors):

In the paper "Dependence of myosin filament structure on intracellular calcium concentration in skeletal muscle" the authors examined the impact of calcium on skinned skeletal muscle and more importantly the movement of the myosin-based reflections. This is a skilled group looking at an extremely important phenomenon (activation of muscle via the thick filament) with the aid of x-ray diffraction. To correct for any issues caused by the diffusion of calcium into the muscle they ingeniously cooled the muscles down to very low temperatures and infused calcium into the muscle once done they quickly warmed the muscles up to 25 degrees Celsius and examined the change in the patterns when calcium was present. While this work is technically challenging, I have some suggestions and questions that need to be addressed.

Major Issues:

1) The point refers to the differences in the patterns observed when going from pCa 9 to pCa 7 and may be due to a misunderstanding in the text. As I understand the text the pCa 9 shots were taken using the same protocol for the "activate" exposures just using the same pCa 9 solution. Yet, as shown in many of these figures, there is a large difference between pCa 9 and the next pCa value (pCa 7) for those traces. Can the authors explain why the 1st MLL, the equatorial reflections, and the meridional reflections are all that different between pCa 9 and pCa 7? Is it possible that changes in the "folded" (I assume Super-Relaxed) heads are occurring at very low levels of calcium (sub 100 nM) which is around the levels seen in resting intact

muscles (~50 nM)? If so, can the authors explain why the SRX (Folded) state would be required during the relaxation step if that phase takes seconds to complete? In the heart of most animals, the beat-to-beat timing is less than one second suggesting that the "refolding" of the heads is not required. What are the authors' opinions on the results of a similar temperature-based experiment done by Malinchik et al (see Malinchik 1997 BiophysJ) also done on rabbit psoas muscles?

2) On page 9 the following text states: "To increase the signal-to-noise on the meridional and layer line reflections, the 1D diffracted intensity profiles for each bundle were divided by their respective I10 values at low calcium to control for variations of the mass in the beam, and profiles for each pCa were averaged between bundles. Background intensity distributions were fitted using a convex hull algorithm and subtracted; the small background remaining when the convex hull algorithm had been used was removed using the intensity from a nearby region of the X-ray pattern containing no reflections or with a linear fit." If that is the case, then why do we see the values that we do? I'm confused as to how the authors came up with the numbers that they did and they need to better clarify this situation since it doesn't make sense with figure 2A.

3) Since the authors observed changes in the lattice spacing with activation, can it be assumed that some of these changes in the reported reflections that occurred were due to sarcomere length changes as mentioned by the authors? Were attempts made to correct for the uncontrolled shortening during activation? The increase of more than 2 nm would equate to a rather sizable shortening in the sarcomere length (~2.15 microns) using the constant lattice volume relationship.

Minor Issues:

1) In a lot of the figures the error bars are hidden behind the symbols, I suggest reporting the values as SD to allow the reader to see them, and/or use open symbols to show what is behind them.

2) It would be helpful to have a side-by-side comparison of the actual images collected to allow the readers to better visualize the changes in the patterns. I suggest putting in the supplemental section 2-by-2 panel with pCa 9.7, 4.5, and whichever submaximal point they want in there to show the changes in meridional and layer line intensities.

3) I had an issue trying to imagine how the setup worked to gather the X-ray data. Could the authors create a schematic representation of the setup so that I and other readers can get a better idea of how the experiment was performed? Maybe include it in a supplemental section.

Reviewer #3 (Comments to the Authors):

This paper describes experiments to detect possible structural changes in vertebrate skeletal thick filaments in response to Ca²⁺, to determine whether Ca²⁺ directly activates thick filaments as suggested recently for cardiac muscle (Ma 2022). Multiple X-ray diffraction signals from thick filaments are recorded at different Ca²⁺ levels and compared with force at these levels to detect possible differences in Ca²⁺-sensitivity. Several signals associated with thick filament activation appear at lower Ca²⁺ levels than force, from which it is concluded that thick filament structure directly responds to Ca²⁺.

The work appears to be carefully carried out, as expected from the known expertise of the authors, and the conclusions seem plausible. But the paper for me was not an easy read. My main concern is that it does not communicate clearly and easily to the non-specialist reader. Many sentences and concepts are quite complex and difficult to follow. It might be more readily absorbed by a specialist, but in my opinion some changes are needed to communicate more widely. Below is a summary of key points that might help, followed by a more detailed list of items suggested for clarification. Please note: referee comments would be much easier to follow if the manuscript lines were numbered.

Key points:

1. p5. "Here we present the results of such a study,...". This sentence leads into the rest of the paper, but it was not clear to me exactly why the proposed experiments were to be done. i.e. what is the purpose of the paper? This is actually stated at top p5, and would be useful to reiterate here so that the reader has in mind why each of the subsequently described experiments was done.

2. In some parts of the paper (e.g. p13, end para1) it seems that a difference of 4 in nH is not considered to be very important, while a difference of say 0.2 in pCa is important. It would help the non-specialist reader to explain the logic here.

3. When a change or difference is described in the text, it would be helpful to show this directly with an arrow (or similar) on the figure itself.

4. After reviewing the careful and intricate experiments carried out by the authors, I am ultimately left feeling disappointed, as there is no clear-cut physical model to explain the results. Maybe it's the nature of the muscle itself, but I would like to come away with some kind of insight into what is going on.

Detailed points:

1. Intro, end para 1. When referring to the folded back structure in vertebrate muscle, refer to Zoghbi 2008 who first showed this.

2. p5 bottom and p6, bottom. Explain logic of protocol? I take it to be: Low temp allows steady state activation by Ca²⁺ without damage to fibers (so there is time for full diffusion of Ca²⁺ into fibers). T jump is rapid and so allows rapid activation of fibers

pre-equilibrated with different Ca²⁺ levels without any issues of Ca²⁺ diffusion. I think this is explained p11, top. Would be useful to have it here.

3. p6, fibre prep. It would be useful to list the compositions of the different solutions here. This would help the reader understand the experimental conditions without having to go back to the references listed under Solutions (p7).

4. p6. What does "skinning solution containing 50% glycerol" mean? Is it skinning solution mixed 50:50 with glycerol (so that dissolved salts are diluted by 50%), or something different?

5. p6. Does glutaraldehyde affect function? How is it prevented from spreading along the fiber?

6. p7. "(between 5 and 0.5 s after the temperature jump increasing calcium from pCa 7 to 4.7)" - not clear what is meant here.

7. p8, para2. Not clear to me exactly what is being done. A cartoon of the setup and procedure would help, or a previous reference. Not clear why it was necessary to remove the specimen from the solution.

8. p10, top. Define M2H, M2L.

9. p11. "we made some control measurements in single demembrated fibres, in which the results from two protocols can be directly compared". It sounds like new steady state experiments are carried out here (to compare with T jump), as would be appropriate. But Fig 1 legend states the data are from Fusi 2016. Please clarify. It would seem to be experimentally more rigorous to directly compare the two methods and not rely on old data.

10. p11, para1. Last sentence implies a comparison of the two protocols is being shown in Table 1. But Table 1 only shows the results from the current T-jump protocol.

11. P12. "The change in the conformation of the myosin motors in single demembrated muscle fibres was measured in the same two protocols" sounds like the measurements using the two protocols are both performed in this paper. But further down this para, it appears the steady state results are obtained from Fusi. Please clarify, and justify this approach if results are taken from the earlier paper.

12. p12. "no significant difference between the corresponding pCa50 and nH values (Table 1)." Text implies the two protocols are compared in Table 1. But this does not seem to be the case (see item 10 above).

13. p13. "As the calcium concentration is increased, the 1,0 reflection becomes weaker and the 1,1 stronger". In fact it seems that the change is not monotonic, as Fig. 2A shows I_{1,0} at pCa7 is stronger than pCa9. What is happening here? pCa 9 seems to be the anomaly.

14. p15. "we conclude that the fraction of myosin motors in the folded conformation is much more sensitive to calcium than force development." Please make explicit the logic. I think the comparison is pCa 6.34 for AML1 with pCa 6.09 for force. If so, please state.

15. p15. "the forbidden reflections (M1, M2, M4 and M5), so-called because they would not be present if the myosins were arranged in a perfect three start helix, have additional components that index on the slightly longer 45.5 nm periodicity (Caremani et al., 2021)". The L-periodicity (45.5 nm) is also discussed on p24, para2, referring back to Caremani 2021 and the possibility that this might arise from heads matching a 45.7 nm titin repeat. Given the recent bioRxiv cryo-EM thick filament structures showing that myosin heads match the 43.0 nm titin repeat, the authors might want to reconsider this discussion point.

16. p15. "All the myosin-based reflections apart from the M6... are weaker...". To my mind M6 also looks weaker in Fig 4.

17. p15. "M6, which is associated with a periodicity in the thick filament backbone, and the M3, which is associated with the periodicity of the myosin motors". This correlation is stated as fact here (and in many papers), with no reference to its origins. It would be very helpful to provide a reference here (and/or explain the logic), so the general reader knows where this comes from. Is there really no significant contribution of myosin heads to the M6?

18. p20. "The same effect was present in other X-ray reflections that are sensitive to the OFF state." Please indicate which figure(s) are being referenced.

19. p24. "We have argued previously calcium activation". I have trouble picturing these concepts. Could they be put into a cartoon?

20. p27. As stated above, I am left wondering exactly what these new, partially-activated, different-periodicity structures might be. I am sure the authors are as well. Do they have any speculation on this, maybe a cartoon as food for thought?

21. p35. Why do the curves shift to the right at higher Ca²⁺ levels?

22. p42. Please define ss and Tj. Also meaning of Force probes?

23. The authors say little about how these structural changes in myosin (whatever they actually are) come about. Do they picture that Ca²⁺ binds to the thick filament. Could Ca²⁺-activated phosphorylation play any role in the changes?

We thank the reviewers for their thorough and detailed analysis and comments, and for their more general perspectives on the presentation of the results and their interpretation. Our response to each comment is listed below with the corresponding changes in the paper specified where relevant. We believe that these changes have significantly improved the clarity of the paper and made it more comprehensible for a wider readership.

Reviewer #1 (Comments to the Authors):

General comments:

In this manuscript by Caremani et al., the authors present some provocative and interesting X-ray diffraction data from skinned skeletal muscle obtained using an uncommon approach, namely temperature jump as a means to rapidly activate skeletal muscle fibers at different pCa values as well as avoid excessive rundown at high temperatures. The methods and data presentation are very clear. The primary findings presented that X-ray signals associated with the perpendicular motors characteristic of isometric force generation had the same calcium sensitivity as force, but X-ray signals associated with perturbations in the folded myosin helix had a much higher calcium sensitivity and that there is a population of myosin motors with a longer axial periodicity than the resting periodicity is seen at low levels of calcium activation, and proposed as an intermediate regulatory state of the myosin motors in the physiological pathway of filament activation. While I have some methodological concerns, the presentation of these findings is clear.

There are a number of awkwardly written and confusing sections in this paper, mostly in the discussion but also in the introduction. For instance, in the abstract and introduction they set things up so that a goal of the paper appears to be to test the hypothesis that there is a direct effect of calcium on structural off to on transitions skeletal muscle, as previously demonstrated in cardiac muscle. Indeed, the title of the paper is "Dependence of myosin filament structure on intracellular calcium concentration in skeletal muscle." In the abstract they say "High-force contraction depends on the release of the folded motors, which can be triggered by stress in the thick filament backbone, but additional mechanisms may link the activation of the thick filaments to that of the thin filaments or to intracellular calcium concentration." In the discussion they argue that their data indicates a need for some other activation mechanism in addition to mechano-sensing but calcium activation per se is discussed only very obliquely in the discussion. Since this was proposed as a guiding hypothesis in the introduction it should be either discussed more directly or set things up differently so it does not appear to be a primary focus of the paper, which I would suggest that it isn't.

Thank you for bringing this fundamental point about the logical structure of the paper to our attention. We have modified the Introduction (page 5) and Discussion (pages 26-29) to address it and to clarify the relationship between those two sections in the light of the Results. As the reviewer suggests at the end of the above paragraph, it was not a primary focus of the paper to 'test the hypothesis that there is a **direct** effect of calcium on structural off to on transitions (in) skeletal muscle' (our emphasis in bold). Our primary aim was to characterise the relationship between the structure of the thick filament and intracellular

calcium concentration ($[Ca]_i$) in the steady-state in near-physiological conditions, as stated in the title of the paper and explained more fully in the abstract. In our view achieving that aim fills a fundamental empirical gap in the muscle literature, because all previously published studies were carried out in conditions in which thick filament-based regulation is not preserved, as explained in the Introduction. The empirical relationship between thick filament structure and $[Ca]_i$ established for the first time in this paper is not specifically linked to a single hypothetical mechanism of thick filament regulation but, because we measured steady force as well as multiple structural parameters, the results do constrain possible mechanisms of thick filament regulation. Those constraints are now summarised more broadly and explicitly in a new section of the Discussion.

There are also concerns with the data themselves. The X-ray data are very lean ($n=2$ for higher calcium measurements) making it difficult to assess the statistical significance of the findings. The choice of 5% dextran to restore the in vivo lattice spacing is also problematic since the authors have shown in previous publications this significantly over-compresses the myofilament lattice to non-physiological lattice spacings (see below). Another concern is the extent to which the OFF-state deficit after a T-jump could be obscuring events that happen at low calcium under more physiological conditions.

These three points are addressed where they come up individually in more detail below.

Specific Concerns

Introduction

The mechanical protocols that were used to establish the role of mechano-sensing to activate thick filaments were confined to a few tens of milliseconds after stimulation, and do not exclude direct effects of calcium on slower time scales.

Moreover, recent evidence suggests that the thick filaments in heart muscle can be directly regulated by calcium (Ma et al., 2022).To test for such a mechanism in skeletal muscle, we determined the dependence of thick filament structure on intracellular free calcium concentration in the steady state.

But the authors don't actually test for a direct calcium effect on thick filament regulation in their experiments. Actin binding was not inhibited in this study so it would not be possible to observe calcium dependent thick filament activation in isolation. Setting it up this way is misleading. What they are observing is the effects of calcium on both thick and thin filament structure together. The best solution perhaps would be to simply discuss calcium dependent thick filament regulation as one of the possible other mechanisms that could supplement mechano-sensing when they discuss the need for such mechanisms.

This point was addressed above under *General Comments*.

Specific Concerns

Introduction.

No previous X-ray study, to our knowledge, has described the calcium dependence of myosin conformation and thick filament structure during active contraction at 25 deg. C in the presence of 5% Dextran.

They need to specify "skeletal muscle" in this sentence. High temperature and 3% dextran was looked at in cardiac muscle eg. Ma et al., 2022 JGP. (paper quoted in this MS).

We added 'of skeletal muscle' to that sentence.

As indicated above, there are good reasons for avoiding 5% dextran. Previous publications by the authors previously showed (Caramini et al. 2021) 5% is substantially more than required to restore in vivo lattice spacing (overshoots by ~2 nm). 3-4% are more commonly used concentrations This needs to be pointed out and its implications discussed.

The probe study of Fusi et al 2016 and the X-ray study of Caremani et al 2021 quoted here used 5% Dextran to compress the filament lattice. Since the present work is based on the results and conclusions of those studies, in particular the demonstration by two independent methods that the OFF state of the thick filament is preserved in those conditions, we retained those previously characterised conditions in the present study. This has the advantage that all the results can be directly compared. Caremani et al 2021 showed that the spacing of the equatorial 1,0 reflection in demembranated fibres from rabbit psoas muscle (the same preparation used here) in the presence of 5% Dextran at 25degC is 1.3nm less than that recorded in intact mouse EDL muscle at the same temperature. This is a reproducible difference, but much smaller than the difference associated with the presence of 5% Dextran in the demembranated fibres, and its significance is in any case limited by the differences in muscle type and species.

Similarly, in the methods: The osmotic agent dextran T500 (5% w/v) (Pharmacia Biotech, Uppsala, Sweden) was added to the experimental solutions to reduce the interfilament spacing to a value similar to that of intact muscle (Linari et al., 2007; Caremani et al., 2019; Caremani et al., 2021)

This is potentially misleading. Depends what you mean by "similar" I don't think they are all that "similar"

We edited this sentence to be quantitative and explicit about the comparison in Caremani et al 2021.

Under these experimental conditions, the activation of the myosin motors on the thick filament in demembranated skeletal muscle fibres is controlled by a stress-dependent mechanism that does not depend on the calcium concentration when active force generation is inhibited by Blebbistatin (Fusi et al., 2016).

I don't think this observation really helps here. Blebbistatin has been showed to favor the closed state of myosin heads in solution studies (Gollapudi et al., 2021 PMID: 34627791) and

sequester myosin heads in a helically ordered state that doesn't reverse quickly. It could be that Blebbistatin stabilized heads are just not sensitive to calcium but are sensitive to strain. Since Blebbistatin stabilized heads may not correspond to a physiological state, I would just omit this sentence here or use it as part of an argument in the discussion where other possible regulatory mechanisms are discussed (see below)

Agreed, this sentence has been removed.

I think the introduction could have been usefully concluded with the main takeaways of the paper that were outlined in the abstract to keep the reader focused on the primary findings.

Done.

Results

Force/pCa done conventionally and with T-jump normalized to Fmax agree reasonably well. Given that the T-jump seems to result in large offsets in the structural measurements, at low calcium it would be desirable to compare the force/unit area tension values

These values are given in the legend of Fig. 1 and are not significantly different.

The X-ray data in Figure 2-5 are pretty lean with only 4 biological replicates for low force points and 2 for the high. I appreciate that it would not be trivial to get more data but we need to be reassured that the conclusions are sound. For instance, in Figure 2 - N=2 for the high force points, so that fit to the I11/I10 curve is not very good. Error bars don't mean much for this low number of n's and nH and pCa50 values are probably not very accurate.

Several of the fits are to summed X-ray data (Fig 2,3, 5) so there is essentially one measurement per point. While the points in these figures relatively show low scatter (compared to figure 2), this makes it very hard to demonstrate significant differences in derived values in Hill fits. Not sure what to make of the error bars on the Hill fits if there are no error bars in the data. While I tend to believe the conclusions that many of the structural changes are more sensitive to calcium than to force, statistically this might be hard to support.

It would have been better to calculate Hill fits for individual preps (replicates), then take the values for all the preps and average them and do stats. This would be much more accurate and confidence inspiring. Obviously, this doesn't work for the summed data.

We have adopted the reviewer's suggestion and fitted Hill curves to the X-ray and force data for individual fibre bundles rather than to the summed data. We then calculated mean +/- SD from this set of fit parameters pCa50 and nH for both force and the X-ray parameters, and those values are shown in the amended version of Table 1. There are some differences compared to the values obtained by fitting the summed data (those presented in the version of Table 1 of the submitted paper) but these are generally small and do not affect any of the conclusions of the submitted paper, although those conclusions are now given a firmer statistical basis. Table 1 now includes statistical tests to compare pCa50 and nH for the X-ray

parameters with those for force. We have also edited Figs 3, 6, 7 and 8 to show the data from individual fibre bundles.

Discussion

*It seems clear that you don't get full restoration of off state after a T-jump at pCa 7. This would seem to complicate interpretation of structural events during activation if a T-jump left heads in a partially ON state. Looking at figure 7 for example it appears that about 1/3 of layer line intensity has gone at the end of the T-jump. In Figure 5 Sm6 indicates that thick filaments about 0.4 % longer than before T-jump. The "structural deficit" in the layer lines and M6 implies a great deal of the initial stages of thick filament activation have already occurred at the end of the T-jump in the absence of calcium. If so, the T-jump might overshadow or possibly blunt activation processes that occur at submaximal calcium. The authors should discuss how does this affects their general conclusions, especially efforts to characterize the transient states indicated by the *state*

This point is the topic of the first section of the Discussion in the submitted paper. There we conclude, as shown by the RLC probe results, that there is a slow phase of the refolding of the motors after the T jump, so indeed the thick filaments are not fully OFF at low [Ca] (e.g. pCa 7) at the time the X-ray measurements are made. However we also showed that the Hill parameters for force and for the probe signals are the same in the T-jump and conventional steady state protocols. Moreover, each X-ray parameter and its associated Hill analysis was carried out using the pCa7 T-jump baseline, where force is still zero. We did not discuss the consequences of the T jump protocol for the * reflection explicitly in this section, but it is present after a T-jump at pCa 7 (Fig. 6A) although not detectable in the steady state at pCa 9 (Fig. 6A and Caremani et al 2021). This is consistent with our conclusion later in the Discussion that the * peak is associated with an intermediate activation state.

"Implications for mechanosensing"

You would a priori expect a sequence of structural events during activation so it is not surprising to me that all the structural events don't all track force. Many of the changes require force generating heads but you would expect the loss of helical ordering to start before significant amount of force is generated. This is in fact, what is seen here.

If myosin activation were directly proportional to force in the steady state, the calcium dependence of structural signals related to filament activation. would be expected to be the same as that of force.

I would expect this only for reflections due to myosin head bound to actin. SM3 increase and IM3 increase should indeed match force which it does.

In contrast, pCa50 for SM6 is much closer to that of force, and nH is the same as that for force within the precision of the measurements. pCa50 for IM3 and AM3 are also close to that of force.

Again this is not unexpected.

"This is clearly not the case (Table 1). pCa_{50} for IML1 or AML1 is larger, by almost 0.3 pCa units, than that for force, as is that for the orientation of the RLC region of the myosin motors (Fig. 1B), which also has a higher value of the Hill coefficient n_H ."

And

" pCa_{50} for the intensities or amplitudes of the forbidden X-ray reflections M1 and M2 that signal regular perturbations in the myosin helix in the relaxed state (Huxley and Brown, 1967) is 0.4-0.5 pCa units larger than that for force, and n_H is again higher."

and

Logically, the loss of helical order would be expected to precede binding so it should also precede force, which the authors show that it does. This is what you expect if there was a calcium-dependent a thick filament-based activation process as shown in cardiac muscle. There may be other possible explanations for this other than direct calcium effects on thick filament structure, but these possibilities need to be discussed. In any case, it seems an odd omission not to discuss what was appeared to be proposed in the introduction as a hypothesis they were testing. The only thing said appears to be:

"It is also possible that the myosin-based structural changes are controlled by a signaling pathway independent of the regulatory calcium sites on troponin. The resolution of those possibilities will require further studies."

Hard to argue with but does not directly address the question.

In this context, the discussion of a non-linear response of activation to force seems a bit forced and there are simpler mechanisms that can be imagined, providing that one accepts that things other than force might be involved in thick filament activation

Taking all the above quotes and comments about mechano-sensing together: Our intention here was to discuss the implications of the present results for the mechano-sensing hypothesis of muscle regulation. Related to the first general comment made by the reviewer, it was not the primary aim of the present experiments to test the mechano-sensing hypothesis, but the results nevertheless have implications for that hypothesis because we measured changes in both force and thick filament structure in response to changes in the independent variable $[Ca]_i$. Most previous tests of the mechano-sensing hypothesis have focused on the dynamics of muscle activation in the first few tens of ms following a fast increase in $[Ca]_i$ as would occur physiologically. The dependence of force and filament structure on $[Ca]_i$ in the steady state, as studied here, is not predictable from those transient responses, not least because we know very little about the structural basis of switching OFF the thick filaments. So the steady state calcium titrations provide new information that might be used to assess or develop the mechano-sensing hypothesis, and that's what we tried to do here.

Perhaps the reviewer means that we devote disproportionate space in the paper to the mechano-sensing hypothesis relative to that devoted to other possible mechanisms of thick filament-based regulation, in line with the reviewer's more general opening comment about

the Discussion. With the hindsight of the reviews, our references to the mechano-sensing hypothesis in the Discussion of the submitted paper as an example of more general implications of the present results were unnecessarily narrow. We have now therefore rewritten this section of the Discussion to present those broader implications for other potential mechanisms of thick filament regulation.

Implications for the physiological regulation of muscle contraction

The above-mentioned concerns about how non-physiological aspects of the T-jump and the high degree of osmotic compression may be obscuring things should be discussed.

The response to these two points has been given above.

General comment:

It is disappointing that there isn't more reference to work by other investigators who have done relevant work.

There are several older X-ray studies using calcium titrations in skinned fibres, and we used the review by Gordon et al (2000) to summarise that older work. We now added a specific reference to the pioneering work of Brenner & Yu (1985) in this regard. Unfortunately no previous published X-ray study using calcium titrations in skinned fibres was carried out at a temperature where thick filament regulation is preserved, as explained in the Introduction.

Reviewer #2 (Comments to the Authors):

In the paper "Dependence of myosin filament structure on intracellular calcium concentration in skeletal muscle" the authors examined the impact of calcium on skinned skeletal muscle and more importantly the movement of the myosin-based reflections. This is a skilled group looking at an extremely important phenomenon (activation of muscle via the thick filament) with the aid of x-ray diffraction. To correct for any issues caused by the diffusion of calcium into the muscle they ingeniously cooled the muscles down to very low temperatures and infused calcium into the muscle once done they quickly warmed the muscles up to 25 degrees Celsius and examined the change in the patterns when calcium was present. While this work is technically challenging, I have some suggestions and questions that need to be addressed.

Major Issues:

1) The point refers to the differences in the patterns observed when going from pCa 9 to pCa 7 and may be due to a misunderstanding in the text. As I understand the text the pCa 9 shots were taken using the same protocol for the "activate" exposures just using the same pCa 9 solution. Yet, as shown in many of these figures, there is a large difference between pCa 9 and the next pCa value (pCa 7) for those traces. Can the authors explain why the 1st MLL, the equatorial reflections, and the meridional reflections are all that different between pCa 9 and pCa 7? Is it possible that changes in the "folded" (I assume Super-Relaxed) heads are occurring at very low levels of calcium (sub 100 nM) which is around the levels seen in resting intact muscles (~50 nM)? If so, can the authors explain why the SRX (Folded) state would be required during the relaxation step if that phase takes seconds to complete? In the

heart of most animals, the beat-to-beat timing is less than one second suggesting that the "refolding" of the heads is not required. What are the authors' opinions on the results of a similar temperature-based experiment done by Malinchik et al (see Malinchik 1997 BiophysJ) also done on rabbit psoas muscles?

This is indeed a misunderstanding; we were not sufficiently clear on this point in the submitted paper. The pCa 9 data were measured in the steady state to give a point of reference to previously published data. The pCa 7 act as the relaxed control for T-jump activation. In the same conditions there is no difference between myosin head conformation between pCa9 and pCa 7. These points are now explained more explicitly on pages 14-15.

Malinchik et al 1997 measured the steady-state dependence of the X-ray pattern from rabbit psoas fibres as a function of temperature in relaxing conditions (ca pCa9). Their results are similar to ours for relaxing solution in the steady state (see also Caremani et al 2021).

2) On page 9 the following text states: "To increase the signal-to-noise on the meridional and layer line reflections, the 1D diffracted intensity profiles for each bundle were divided by their respective I10 values at low calcium to control for variations of the mass in the beam, and profiles for each pCa were averaged between bundles. Background intensity distributions were fitted using a convex hull algorithm and subtracted; the small background remaining when the convex hull algorithm had been used was removed using the intensity from a nearby region of the X-ray pattern containing no reflections or with a linear fit." If that is the case, then why do we see the values that we do? I'm confused as to how the authors came up with the numbers that they did and they need to better clarify this situation since it doesn't make sense with figure 2A.

We were not sufficiently clear in this paragraph, which refers only to the meridional and layer line reflections, not the equatorial reflections in Fig. 2. We now made this explicit in the text. Note that in response to comments from reviewer 1, we have now extended the analysis of data from individual fibre bundles to give a measure of biological reproducibility.

3) Since the authors observed changes in the lattice spacing with activation, can it be assumed that some of these changes in the reported reflections that occurred were due to sarcomere length changes as mentioned by the authors? Were attempts made to correct for the uncontrolled shortening during activation? The increase of more than 2 nm would equate to a rather sizable shortening in the sarcomere length (~2.15 microns) using the constant lattice volume relationship.

We agree that the changes in d10 in Fig. 2C are likely to be associated with sarcomere shortening. The latter is likely to be larger in the fibre bundles than in single fibres, as discussed on page 22, and the total sarcomere shortening at full calcium activation could be from 2.4 to about 2.2um, as suggested by the reviewer. This is the plateau region of the force-sarcomere length relationship, so the shortening is not expected to have a significant effect on force at maximum activation, but may reduce the co-operativity of the force-pCa relation, as also discussed on page 22.

Minor Issues:

1) In a lot of the figures the error bars are hidden behind the symbols, I suggest reporting the values as SD to allow the reader to see them, and/or use open symbols to show what is behind them.

In the submitted paper, only mean values were shown for the meridional and layer line reflections. In response to comments from reviewer 1, we have edited Figs 3, 6, 7 and 8 to show directly the data from individual fibre bundles.

2) It would be helpful to have a side-by-side comparison of the actual images collected to allow the readers to better visualize the changes in the patterns. I suggest putting in the supplemental section 2-by-2 panel with pCa 9,7, 4.5, and whichever submaximal point they want in there to show the changes in meridional and layer line intensities.

We previously published such images obtained from the same preparation and at the same beamline showing the effects of cooling and lattice compression (Caremani et al. 2021); those images allow the quality of the diffraction patterns to be assessed. Unfortunately the quadrant image with 4 selected pCas suggested by the reviewer does not show the different calcium sensitivities clearly. We chose the radial and axial integrations and individual reflection intensities and spaces to show the results in Figs 2-8 because the combination of background subtraction and integration and the simultaneous comparison of all the different pCas for each parameter makes the results much clearer.

3) I had an issue trying to imagine how the setup worked to gather the X-ray data. Could the authors create a schematic representation of the setup so that I and other readers can get a better idea of how the experiment was performed? Maybe include it in a supplemental section.

We have now included such a diagram as the new Fig. S1.

Reviewer #3 (Comments to the Authors):

This paper describes experiments to detect possible structural changes in vertebrate skeletal thick filaments in response to Ca²⁺, to determine whether Ca²⁺ directly activates thick filaments as suggested recently for cardiac muscle (Ma 2022). Multiple X-ray diffraction signals from thick filaments are recorded at different Ca²⁺ levels and compared with force at these levels to detect possible differences in Ca²⁺-sensitivity. Several signals associated with thick filament activation appear at lower Ca²⁺ levels than force, from which it is concluded that thick filament structure directly responds to Ca²⁺.

The work appears to be carefully carried out, as expected from the known expertise of the authors, and the conclusions seem plausible. But the paper for me was not an easy read. My main concern is that it does not communicate clearly and easily to the non-specialist reader. Many sentences and concepts are quite complex and difficult to follow. It might be more readily absorbed by a specialist, but in my opinion some changes are needed to communicate more widely. Below is a summary of key points that might help, followed by a more detailed list of items suggested for clarification. Please note: referee comments would be much easier to follow if the manuscript lines were numbered.

We address the key points individually below. We did not intend to convey the conclusion that thick filament structure responds **directly** to Ca^{2+} , so we were not sufficiently clear. We have therefore rewritten the relevant section of the Discussion to make the conclusions more explicit and hopefully clearer, as also requested by reviewer 1.

Key points:

1. p5. "Here we present the results of such a study,...". This sentence leads into the rest of the paper, but it was not clear to me exactly why the proposed experiments were to be done. i.e. what is the purpose of the paper? This is actually stated at top p5, and would be useful to reiterate here so that the reader has in mind why each of the subsequently described experiments was done.

We added a sentence on pages 5-6 to clarify the aims and followed it by a brief summary of the conclusions (also requested by reviewer 1).

2. In some parts of the paper (e.g. p13, end para1) it seems that a difference of 4 in nH is not considered to be very important, while a difference of say 0.2 in pCa is important. It would help the non-specialist reader to explain the logic here.

Some of the nH values in Table 1 have large SDs, of the order of 1 or larger, in marked contrast with the typical SD for pCa50, which is around 0.05-0.1. Moreover nH is larger in fibre bundles that have been activated in air than in single fibres in solution, as discussed on page 22. However nH values for the intensities of the forbidden reflections and the troponin reflection do seem to be significantly higher than those for force and the intensity of the M3 reflection for example. A higher value of nH would suggest a more co-operative structural change, as might be expected for a structural transition involving multiple intermolecular interactions between myosin motors. This point is now made explicit on page 26.

3. When a change or difference is described in the text, it would be helpful to show this directly with an arrow (or similar) on the figure itself.

That works well for molecular structure Figs, but seems to us not helpful for the 2D variable relationships in the Figs in this paper, where we follow the field-standard convention of points/error bars for the data and lines for the fit, and using colour-coding to identify different conditions, with key insets to the colours in the Figs where necessary backed up textually in the legends. An arrow would necessarily be local and limited, detracting from our focus on the overall relationships that summarise the contribution of the multiplicity of data points that contribute to each result.

4. After reviewing the careful and intricate experiments carried out by the authors, I am ultimately left feeling disappointed, as there is no clear-cut physical model to explain the results. Maybe it's the nature of the muscle itself, but I would like to come away with some kind of insight into what is going on.

We think that this is the consequence of the current state of knowledge about muscle structure and function, combined with the partial nature of the information that can be

obtained by X-ray diffraction in the absence of a comprehensive real-space structural model for the muscle filaments. However we do think that the results constrain possible mechanisms of thick filament regulation, and have now added a new section to the Discussion to describe those constraints in detail, as also requested by reviewer 1.

Detailed points:

1. Intro, end para 1. When referring to the folded back structure in vertebrate muscle, refer to Zoghbi 2008 who first showed this.

Done

2. p5 bottom and p6, bottom. Explain logic of protocol? I take it to be: Low temp allows steady state activation by Ca²⁺ without damage to fibers (so there is time for full diffusion of Ca²⁺ into fibers). T jump is rapid and so allows rapid activation of fibers pre-equilibrated with different Ca²⁺ levels without any issues of Ca²⁺ diffusion. I think this is explained p11, top. Would be useful to have it here.

Done

3. p6, fibre prep. It would be useful to list the compositions of the different solutions here. This would help the reader understand the experimental conditions without having to go back to the references listed under Solutions (p7).

Done.

4. p6. What does "skinning solution containing 50% glycerol" mean? Is it skinning solution mixed 50:50 with glycerol (so that dissolved salts are diluted by 50%), or something different?

The full composition of the skinning and storage solutions is now given.

5. p6. Does glutaraldehyde affect function? How is it prevented from spreading along the fiber?

One small drop of glutaraldehyde is applied with a small tube (internal diameter 0.6 mm) on the trabecular end next to the clip, in air after cooling at 2degC. This prevents spread along the fiber, so the central region where X-ray measurements are made has not been in contact with glutaraldehyde. The method is described in Linari et al. 2007.

6. p7. "(between 5 and 0.5 s after the temperature jump increasing calcium from pCa 7 to 4.7)" - not clear what is meant here.

This has been reworded for clarity.

7. p8, para2. Not clear to me exactly what is being done. A cartoon of the setup and procedure would help, or a previous reference. Not clear why it was necessary to remove the specimen from the solution.

Fibre bundles were removed from solution for the X-ray exposures to minimise scattering from the solution and the windows of the chamber. A diagram of the set-up has been added as Fig. S1.

8. p10, top. Define M2H, M2L.

Done

9. p11. "we made some control measurements in single demembrated fibres, in which the results from two protocols can be directly compared". It sounds like new steady state experiments are carried out here (to compare with T jump), as would be appropriate. But Fig 1 legend states the data are from Fusi 2016. Please clarify. It would seem to be experimentally more rigorous to directly compare the two methods and not rely on old data.

The order parameters from the bifunctional probes are extremely reproducible between fibres, as can be seen from the standard errors in Fig. 1B. This is fundamentally because they are calculated from ratios of polarised components with extremely small instrumental errors, but also because the rabbit psoas fibres have very low biological variability in this parameter (including variability due to different animals, preps and solutions). The use of data from Fusi et al is now made explicit in the text as well as in the Fig. 1 legend.

10. p11, para1. Last sentence implies a comparison of the two protocols is being shown in Table 1. But Table 1 only shows the results from the current T-jump protocol.

This is incorrect; Table 1 shows both T-jump (TJ) and steady state (SS) data. This is now made explicit in the Table 1 legend.

11. P12. "The change in the conformation of the myosin motors in single demembrated muscle fibres was measured in the same two protocols" sounds like the measurements using the two protocols are both performed in this paper. But further down this para, it appears the steady state results are obtained from Fusi. Please clarify, and justify this approach if results are taken from the earlier paper.

OK, this sentence has been clarified as requested.

12. p12. "no significant difference between the corresponding pCa50 and nH values (Table 1)." Text implies the two protocols are compared in Table 1. But this does not seem to be the case (see item 10 above).

The two protocols (SS and TJ) are compared in Table 1, see above.

13. p13. "As the calcium concentration is increased, the 1,0 reflection becomes weaker and the 1,1 stronger". In fact it seems that the change is not monotonic, as Fig. 2A shows I1,0 at pCa7 is stronger than pCa9. What is happening here? pCa 9 seems to be the anomaly.

The ratio of the intensities of the 1,1 and 1,0 reflections is a more reliable parameter, and increases monotonically with calcium concentration. The absolute intensity of the 1,0 reflection can vary slightly depending on the position on the bundle. The difference in the equatorial reflections between pCa 9 and pCa 7 is due to the use of the temperature jump protocol rather than the difference in pCa, as shown by the control experiments in Fig. 1. This point was also raised by reviewer 2, and we have clarified the relevant text on pages 14-15.

14. p15. "we conclude that the fraction of myosin motors in the folded conformation is much more sensitive to calcium than force development." Please make explicit the logic. I think the comparison is pCa 6.34 for AML1 with pCa 6.09 for force. If so, please state.

Yes, we now made this explicit on pp. 15-16.

15. p15. "the forbidden reflections (M1, M2, M4 and M5), so-called because they would not be present if the myosins were arranged in a perfect three start helix, have additional components that index on the slightly longer 45.5 nm periodicity (Caremani et al., 2021)". The L-periodicity (45.5 nm) is also discussed on p24, para2, referring back to Caremani 2021 and the possibility that this might arise from heads matching a 45.7 nm titin repeat. Given the recent bioRxiv cryo-EM thick filament structures showing that myosin heads match the 43.0 nm titin repeat, the authors might want to reconsider this discussion point.

We have incorporated this point into the Discussion starting on pages 24-25, in which these recent pre-prints are referenced.

16. p15. "All the myosin-based reflections apart from the M6... are weaker...". To my mind M6 also looks weaker in Fig 4.

Width-corrected I_{M6} (Fig. S2) is indeed slightly stronger at full activation than in relaxation. Fig. 4 is not width-corrected because different meridional reflections may have different radial widths.

17. p15. "M6, which is associated with a periodicity in the thick filament backbone, and the M3, which is associated with the periodicity of the myosin motors". This correlation is stated as fact here (and in many papers), with no reference to its origins. It would be very helpful to provide a reference here (and/or explain the logic), so the general reader knows where this comes from. Is there really no significant contribution of myosin heads to the M6?

The most compelling albeit indirect evidence comes from the almost constant intensity of the M6 reflection in response to a rapid length step applied during contraction, which produces a large change in the intensity of the M3 reflection that can be understood in terms of a change in conformation of actin-attached myosin motors (Reconditi et al 2004; Huxley et al 2006). These references have been added on page 17. In general both the spacings and the intensities of the M3 and M6 appear to be uncoupled in many published protocols and muscle preparations, as well as in the present ones. The structural basis of this uncoupling is not understood, but may become clearer in the light of emerging cryo-EM evidence about the detailed structure of the C-zone of the thick filaments.

18. p20. *"The same effect was present in other X-ray reflections that are sensitive to the OFF state." Please indicate which figure(s) are being referenced.*

Done.

19. p24. *"We have argued previously calcium activation". I have trouble picturing these concepts. Could they be put into a cartoon?*

We believe that this would be premature, but the concepts referred to here are now made more general in the new Discussion section in the context of three alternative, but not mutually exclusive, mechanisms of thick filament regulation.

20. p27. *As stated above, I am left wondering exactly what these new, partially-activated, different-periodicity structures might be. I am sure the authors are as well. Do they have any speculation on this, maybe a cartoon as food for thought?*

As noted in response to the previous point, we prefer to discuss these concepts in the text, giving equal consideration to function as well as structure, in the context of three alternative, but not mutually exclusive, mechanisms of thick filament regulation.

21. p35. *Why do the curves shift to the right at higher Ca²⁺ levels?*

This is because of the contribution of the actin-based AL1 layer line with shorter helical periodicity.

22. p42. *Please define ss and Tj. Also meaning of Force probes?*

Done.

23. *The authors say little about how these structural changes in myosin (whatever they actually are) come about. Do they picture that Ca²⁺ binds to the thick filament. Could Ca²⁺-activated phosphorylation play any role in the changes?*

This is the subject of the new section in the Discussion. Skinned fibres from rabbit psoas muscle have low levels of RLC phosphorylation even at high [Ca], although the RLC can be phosphorylated if exogenous kinase and calmodulin are added.

August 17, 2023

Dr. Elisabetta Brunello
King's College London
Randall Centre for Cell and Molecular Biophysics
New Hunt's House, Guy's Campus
London SE1 1UL
United Kingdom

Re: 202313393R1

Dear Elisabetta,

I am pleased to let you know that your manuscript, entitled "Dependence of myosin filament structure on intracellular calcium concentration in skeletal muscle" is scientifically acceptable for publication in Journal of General Physiology. Formal acceptance will follow when it is modified in accordance with the referees' few remaining remarks (see below) and our editorial policies.

Please note items that need attention are listed at the bottom of this email (under 'manuscript formatting checklist') and on the attached marked-up pdf file. Please also be sure to include a letter addressing the reviewers' comments point-by-point (if applicable) and a copy of the text with alterations highlighted (boldfaced or underlined). Your manuscript should be a double-spaced MS Word file and include editable tables, if appropriate.

JGP now requires a data availability statement for all research article submissions. These statements will be published in the article directly above the Acknowledgments. The statement should address all data underlying the research presented in the manuscript. Please visit the JGP instructions for authors for guidelines and examples of statements at <https://rupress.org/jgp/pages/editorial-policies#data-availability-statement>.

Lastly, JGP adds short captions to articles listed on our weekly newest article emails. If you haven't, please provide a short, ~40-word summary statement for the online JGP table of contents and alerts. This summary should describe the context and significance of the findings for a general readership and be placed on/near the title page.

Please submit your final files via this link:
Link Not Available

Thank you for choosing to publish your research in JGP and please feel free to contact me with any questions.

Sincerely,

Henk L. Granzier, Ph.D.
On behalf of Journal of General Physiology

Journal of General Physiology's mission is to publish mechanistic and quantitative molecular and cellular physiology of the highest quality; to provide a best in class author experience; and to nurture future generations of independent researchers.

Manuscript formatting checklist:

- MS Word document of text needed (including editable tables)
- MS Word document of supplemental text needed, if applicable (including figure legends and editable tables)
- Brief Statement describing supplementary information needed, if applicable (in subsection at end of Materials & Methods)
- Please include a data availability statement preceding the Acknowledgments section. Please see <https://rupress.org/jgp/pages/editorial-policies#data-availability-statement>
- Figures created at sufficient resolution and in acceptable format (including supplemental if applicable). If working in Illustrator, we prefer .ai or .eps file format. If working in Photoshop please use 600dpi/1000dpi .tiff or .psd file format. Minimum resolution at estimated print size: Minimum resolution for all figures is 600 dpi. For figures that contain both photographs and line art or text, 600 dpi is highly recommended. Figures containing only black and white elements (line art, no color, and no gray) should be 1,000 dpi. Maximum figure size is 7 in wide x 9 in high (17.5 x 22.8 cm) at the correct resolution. <https://jgp.rupress.org/fig-vid-guidelines>
- Supplemental figures, if any, conforming to same guidelines as manuscript figures (noted above)
- If images resemble one from a prior publications, the author must seek permissions (to reproduce or adapt) from the original publisher. [You can resubmit your paper while waiting to hear back from the original publisher but please keep us updated]

- All authors must complete a disclosure form prior to acceptance. A link to complete the form has been sent to all coauthors. Please provide the editorial office with updated email addresses if necessary

Reviewer #1 (Comments to the Authors):

The revised manuscript is much improved and with one small change is in my opinion acceptable for publication.

The change involves this statement in the discussion:

"For example, increasing [Ca²⁺] in the range pCa 9 to pCa 4.5 abolished the M3 and M6 reflections in that study, even in control conditions, whereas here the intensity of the M3 reflection increased dramatically with increasing"

Looking at figure 3 of Ma et al 2022 it does indeed appear that IM3 and IM6 go to zero at pCa 4.5. If you look at the text of Ma et al., 2022, however, it says

"Compared to pCa 8, the IMLL1 and IM3 intensities at pCa 4.5 decrease to 27 {plus minus} 2.5% and 35 {plus minus} 1.9%, respectively, in the inhibitor group, whereas it decreases to 18 {plus minus} 1.6% and 26 {plus minus} 2.4% in the control group (inset in Fig. 3, a and b)."

The contradiction between porcine cardiac and rabbit skeletal muscle is still there, you just need to replace "abolished" with "greatly reduced" and also indicate that that the two experiments were done on very different muscle systems. It is probably unwise to try to extrapolate from cardiac to skeletal in general.

Reviewer #2 (Comments to the Authors):

Response to the Author's comments:

After reading the comments of the other reviewers and the author's comments/changes to the manuscript I feel that the authors have made significant gains in their paper but still have some work to do. With the evidence that these concerns are addressed, I then suggest that this paper be accepted for publication.

Major Concerns

1) I'm still confused about what the authors are trying to convey in the methods section. The data in Figure 3 would suggest that the heads are in a strongly folded helical states since the intensity of that peak is much greater than any of the other peaks. Furthermore, when 100 nM Free Calcium is added you get a rather sizable drop in the intensity (suggesting heads are moving out of the folded state). What I'm still not clear on is what is causing this initial movement; is it the addition of a small amount of calcium or the temperature change brought on by the temperature jumps? As they have stated in Figure 1 there is a rather large wait for the heads to return to "normal" once they are warmed back up. Can the authors clarify how long after the temperature jump did they wait before taking the shot?

2a) Page 10 the line "The 1D diffracted intensity profiles of the meridional and layer line reflections for each bundle were divided by their respective I₁₀ values at low calcium concentration to control for variations of the mass in the beam" please define which low calcium concentration value was used. As you stated in many sections of the paper, the 1,0 reflections change (get lower) when calcium is increased so using the 1,0 from any of the other calcium concentrations can lead to errors. The better thing to do is to use the intensity of the pCa 9 shot since then the heads are not "moving" and the system is closest to intact relaxed.

2b) On that same idea, based on the comments of reviewer 3 and your response, on the figures where the intensity plot profiles are shown (Figures 2A, 3A, and 4) were they corrected for box width? Knowing that the width of the peaks can vary from shot to shot it's best to get the entire peaks in one plot profile and then correct for box widths. That might reduce some of the confusion and also make including Figure S2 irrelevant since we would then be able to see the increase in intensity.

Minor Concerns

1) It may be prudent to leave out the data from the pCa 9 set since you have large changes in most structural parameters, either that or be very clear in stating that the pCa 9 data didn't go through a temperature jump like the other data points and that you took the exposures after a certain time frame presumably enough time for the heads to "recover".

Reviewer #3 (Comments to the Authors):

The paper has been improved considerably in response to reviewer comments, including improvements in clarity and accessibility. I like the revised discussion and inclusion of reference to recent thick filament cryo-EM data.

We thank the reviewers for their additional comments. Our response to each comment is listed below with the corresponding changes in the paper specified where relevant.

Reviewer #1 (Comments to the Authors):

The revised manuscript is much improved and with one small change is in my opinion acceptable for publication.

The change involves this statement in the discussion:

"For example, increasing [Ca²⁺] in the range pCa 9 to pCa 4.5 abolished the M3 and M6 reflections in that study, even in control conditions, whereas here the intensity of the M3 reflection increased dramatically with increasing"

Looking at figure 3 of Ma et al 2022 it does indeed appear that IM3 and IM6 go to zero at pCa 4.5. If you look at the text of Ma et al., 2022, however, it says

"Compared to pCa 8, the IM11 and IM3 intensities at pCa 4.5 decrease to 27 {plus minus} 2.5% and 35 {plus minus} 1.9%, respectively, in the inhibitor group, whereas it decreases to 18 {plus minus} 1.6% and 26 {plus minus} 2.4% in the control group (inset in Fig. 3, a and b)."

The contradiction between porcine cardiac and rabbit skeletal muscle is still there, you just need to replace "abolished" with "greatly reduced" and also indicate that that the two experiments were done on very different muscle systems. It is probably unwise to try to extrapolate from cardiac to skeletal in general.

We have replaced "abolished" with "greatly reduced".

Reviewer #2 (Comments to the Authors):

Response to the Author's comments:

After reading the comments of the other reviewers and the author's comments/changes to the manuscript I feel that the authors have made significant gains in their paper but still have some work to do. With the evidence that these concerns are addressed, I then suggest that this paper be accepted for publication.

Major Concerns

1) I'm still confused about what the authors are trying to convey in the methods section. The data in Figure 3 would suggest that the heads are in a strongly folded helical states since the intensity of that peak is much greater than any of the other peaks. Furthermore, when 100 nm Free Calcium is added you get a rather sizable drop in the intensity (suggesting heads are moving out of the folded state). What I'm still not clear on is what is causing this initial movement; is it the addition of a small amount of calcium or the temperature change brought on by the temperature jumps? As they have stated in Figure 1 there is a rather large wait for the heads to return to "normal" once they are warmed back up. Can the authors clarify how long after the temperature jump did they wait before taking the shot?

The pCa 9 data were measured in the steady state to give a point of reference to previously published data. The pCa 7 act as the relaxed control for T-jump activation. X-ray signals were acquired ~1s after the T-jump, therefore the difference between pCa 9 and 7 is due to the temperature effect as explained in Figure 1. The time of the X-ray exposure has now been added to the Methods.

2a) Page 10 the line "The 1D diffracted intensity profiles of the meridional and layer line reflections for each bundle were divided by their respective I10 values at low calcium concentration to control for variations of the mass in the beam" please define which low calcium concentration value was used. As you stated in many sections of the paper, the 1,0 reflections change (get lower) when calcium is increased so using the 1,0 from any of the other calcium concentrations can lead to errors. The better thing to do is to use the intensity of the pCa 9 shot since then the heads are not "moving" and the system is closest to intact relaxed.

Data were normalized by pCa9 at 25C. We have now made this explicit in Methods.

2b) On that same idea, based on the comments of reviewer 3 and your response, on the figures where the intensity plot profiles are shown (Figures 2A, 3A, and 4) were they corrected for box width? Knowing that the width of the peaks can vary from shot to shot it's best to get the entire peaks in one plot profile and then correct for box widths. That might reduce some of the confusion and also make including Figure S2 irrelevant since we would then be able to see the increase in intensity.

Figure 2A and 3A were not corrected by the cross-meridional width because they are equatorial and layer line reflections, respectively. Figure 4 is not corrected by the cross-meridional width, because it could be different for each meridional reflection.

Minor Concerns

1) It may be prudent to leave out the data from the pCa 9 set since you have large changes in most structural parameters, either that or be very clear in stating that the pCa 9 data didn't go through a temperature jump like the other data points and that you took the exposures after a certain time frame presumably enough time for the heads to "recover".

The pCa 9 data were measured in the steady state to give a point of reference to previously published data.

Reviewer #3 (Comments to the Authors):

The paper has been improved considerably in response to reviewer comments, including improvements in clarity and accessibility. I like the revised discussion and inclusion of reference to recent thick filament cryo-EM data.

Thank you very much.